# Deep Submodular Peripteral Networks

**Gantavya Bhatt**[*]     **Arnav M. Das**[*]     **Jeffrey A. Bilmes**
University of Washington, Seattle, WA 98195
{gbhatt2, arnavmd2, bilmes}@uw.edu

## Abstract

Submodular functions, crucial for various applications, often lack practical learning methods for their acquisition. Seemingly unrelated, learning a scaling from oracles offering graded pairwise preferences (GPC) is underexplored, despite a rich history in psychometrics. In this paper, we introduce deep submodular peripteral networks (DSPNs), a novel parametric family of submodular functions, and methods for their training using a GPC-based strategy to connect and then tackle both of the above challenges. We introduce newly devised GPC-style "peripteral" loss which leverages numerically graded relationships between pairs of objects (sets in our case). Unlike traditional contrastive learning, or RHLF preference ranking, our method utilizes graded comparisons, extracting more nuanced information than just binary-outcome comparisons, and contrasts sets of any size (not just two). We also define a novel suite of automatic sampling strategies for training, including active-learning inspired submodular feedback. We demonstrate DSPNs' efficacy in learning submodularity from a costly target submodular function and demonstrate its superiority both for experimental design and online streaming applications.

## 1 Introduction and Background

This paper jointly addresses two seemingly disparate problems presently open in the machine learning community.

The first is that of identifying a useful practical scalable submodular function that can be used for real-world data science tasks. Submodular functions, set functions that exhibit a diminishing returns property (see Appendix B), have received considerable attention in the field of machine learning. This has fostered new algorithms that offer near-optimal solutions for various applications. These applications include detecting disease outbreaks [56], modeling fine structure in computer vision [41], summarizing images [102, 72, 26, 37], active learning [34, 32, 25], compressed sensing, structured convex norms, and sparsity [4, 24], fairness [17, 43], efficient model training [57, 58, 70, 69] recommendation systems [71], causal structure [114, 97], and brain parcellating [85] (see also the reviews [101, 12]). Despite these myriad applications, most research on submodularity has been on the algorithmic and theoretical side which assumes the submodular function needing to be optimized is already at hand. While there has been work showing the theoretical challenges associated with learning submodularity [5, 7, 6], there is relatively little work on practical methods to produce a useful submodular function in the first place (a few exceptions include [60, 92, 103]). Often, because it works well, one uses the non-parametric submodular facility location (FL) function, i.e., for $A \subseteq V$ with $|V| = n$, $f(A) = \sum_{i=1}^{n} \max_{a \in A} s_{a,i}$ where $s_{a,i} \geq 0$ is a similarity between items $a$ and $i$. On the other hand, the FL function's computational and memory costs grow quadratically with dataset size $n$ and so FL can become infeasible for large data or online streaming applications. There is a need for practical strategies to obtain useful, scalable, general-purpose, and widely applicable submodular functions.

---

[*]Equal Contribution

38th Conference on Neural Information Processing Systems (NeurIPS 2024).

The second problem we address in this paper is the following: how best, in a modern machine-learning context, can one learn a scaling from oracles that, for a given query, each provide numerically graded pairwise comparisons between two choices? That is, given two choices, $A$ and $B$, an oracle provides a score $Score(A, B) \in \mathbb{R}$ which is positive if $A$ is preferred to $B$, negative if $B$ is preferred to $A$, zero if indifferent, and where the absolute value provides the degree of preference $A$ or $B$ has over the other. The oracle could be an individual human annotator, or could be a combinatorially expensive desired target function — in either case we assume that it is infeasible to optimize over the oracle but it is feasible to query. Learning a "scaling" means learning a scalar-valued function $f$ that respects these graded pairwise scores in that $f(A) - f(B) \approx Score(A, B)$.

A special case of such preference elicitation from human oracles have been studied going back many decades. For example, the psychometric "Law of comparative judgment" [99] focuses on establishing numeric interval scales based on knowing preferences between pairs of choices (e.g., $A$ vs $B$). The Bradley-Terry [14] model (generalized to ranking elicitations in the Luce-Shephard model [65, 90]) also involves preferences among elements of pairs. These preferences, however, are not graded and rather are either only binary ($A \prec B$ or $B \prec A$), ternary (allowing also for indifference), or quaternary (allowing also for incomparability). Thus, each oracle query provides minimal information about a pair. Therefore, multiple preference ratings are often aggregated where a collection of (presumed random i.i.d.) oracles vote on the preference between items of the same pair. This produces a histogram of preference ratings $C_{ij}$ (a count of how many times $i$ is preferred to $j$ [91]) and this can be used in modern reinforcement learning with human feedback (RLHF) systems [109, 19, 76, 111, 105] for reward learning (i.e., inverse reinforcement learning [3, 1]). Overall, the above modeling strategies are known in the field of psychometric theory [33] as "paired comparisons" where it is said that "paired comparison methods generally give much more reliable results" [74] for models of human preference expression and elicitation.

There are other models of preference elicitation besides paired comparisons. We explore "graded pairwise comparisons" (GPC) which also have been studied for quite some time [87, 9, 10]. Quite recently, it was found that "GPCs are expected to reduce faking compared with Likert-type scales and to produce more reliable, less ipsative trait scores than traditional-binary forced-choice formats" [61] such as paired comparisons. An intuitive reason for this is that, once we get to the point that an oracle is asked only if $A \prec B$ or $B \prec A$, this does not elicit as much information out of the oracle as would asking for $Score(A, B)$ even though the incremental oracle effort for the latter is often negligible. As far as we know, despite GPC's potential information extraction efficiency advantages, GPC has not been addressed in the modern machine learning community. Also, while histograms $C_{i,j}$ are appropriate for maximum likelihood estimation (MLE) of non-linear logistic and softmax style regression [19, 76, 115, 111], MLE with a Bradley-Terry style model is suboptimal to learn scoring from graded preferences $Score(A, B) \in \mathbb{R}$ which can be an arbitrary signed real number.

In the present paper we simultaneously address both of the above concerns via the introduction and training of **deep submodular peripteral networks** (DSPNs). In a nutshell, DSPNs are an expressive theoretically interesting parametric family of submodular functions that can be successfully learnt using a new GPC-style loss that we introduce below. In our work, since the teacher will be an expensive FL function and the learner is a DSPN set function, we can view this as a form of knowledge distillation. Our offering, therefore, is very far from incremental — rather, we introduce a new provably more expressive model family (DSPNs), and a new learning paradigm (GPC-style learning) for distilling an expensive FL set function down to an expressive parametric learner.

An outline of our paper follows. In §2 we define the class of DSPN functions, motivating the name "peripteral", and offering theoretical comparisons with the class of deep sets [110]. Next, §3 describes a new GPC-style "peripteral" loss that is appropriate to train a DSPN but can be used in any GPC-based preference elicitation learning task. For DSPN learning, this happens by producing a list of pairs of subsets $E, M$ that can be used to transfer information from the oracle target to the DSPN being learnt. §4 describes several $E, M$ pair sampling strategy including an active-learning inspired submodular feedback approach. §5 empirically evaluates DSPN learning based on how effectively information is transferred from the target oracle and performance on downstream tasks such as experimental design. We find that the peripteral loss outperforms other losses for training DSPNs, and show that the DSPN architecture is more effective at learning from the target function than baseline methods such as Deep Sets and SetTransformers. We also find that the grading in GPC indeed improves performance over binary-only comparisons.

While the above introduction situates our research and establishes the setting of our paper in the context of previous related work, our relation to and advancement over **other related work** is fully explored in our Appendix in §A. The appendices continue (starting at §B) offering further details of DSPN novelty (§C) and learning via the GPC-style peripteral loss.

## 2  Deep Submodular Peripteral Networks

Before starting this section, it may assist the reader to consider our brief overview of submodularity, matroids, and weighted matroid rank in §B, and a brief review of deep submodular functions in §C, both provided in the appendix.

A deep submodular peripteral network (DSPN) is a new expressive trainable parametric nonsmooth subdifferentiable submodular function. A DSPN has three stages: (1) a "pillar" stage; (2) a submodular-preserving permutation-invariant aggregation stage; and (3) a "roof" stage. When considered together, these three stages resemble an ancient Greek or Roman "peripteral" temple as shown in Figures 1 and 10. In this paper, all DSPNs are also monotone non-decreasing and normalized (see §B). As a set function $f : 2^V \to \mathbb{R}_+$, a DSPN maps any subset $A \subseteq V$ to a non-negative real number. We consider sets as object indices so $A$ might be a set of image indices while $\{x_i : i \in A\}$ are the objects being referred to. Hence, w.l.o.g., $V = [n]$.

The first stage of a DSPN consists of a set of $|A|$ pillars one for each element in a set $A$ that is being evaluated. The number of pillars changes depending on $|A|$ the size of the set $A$ being evaluated. Each pillar is a many-layered deep neural structure $\phi_{w_\mathfrak{p}} : \mathcal{D}_X \to \mathbb{R}_+^d$, parameterized by the vector $w_\mathfrak{p}$ of real values, that maps from an object (from domain $\mathcal{D}_X$ which could be an image, string, etc.) to a non-negative embedded representation of that object. For any $v \in V$ and object $x_v$, then $\phi_{w_\mathfrak{p}}(x_v)$ is a $d$-dimensional non-negative real-valued vector $(\phi_{w_\mathfrak{p}}(x_v)_1, \phi_{w_\mathfrak{p}}(x_v)_2, \ldots, \phi_{w_\mathfrak{p}}(x_v)_d) \in \mathbb{R}_+^d$. The parameters $w_\mathfrak{p}$ are shared for all objects. Also, $\phi_{w_\mathfrak{p}}(x_:)_j \in \mathbb{R}_+^n$ refers, for any $j \in [d]$, to the $n$-dimensional vector of the $j^{\text{th}}$ pillar outputs for all $v \in V$.

The second stage of a DSPN is a submodular preserving and permutation-invariant aggregation. Before describing this generally, we start with a simple example. Each element $\phi_{w_\mathfrak{p}}(x_a)_j$ of $\phi_{w_\mathfrak{p}}(x_a)$ is a score for $x_a$ along dimension $j$ and so a non-negative modular set function can be constructed via $m_j(A) = \sum_{a \in A} \phi_{w_\mathfrak{p}}(x_a)_j$. Such a summation is an aggregator that, along dimension $j$ simply sums up the $j^{\text{th}}$ contribution for every object in $A$. A simple way to convert this to a monotone non-decreasing submodular function composes it with a non-decreasing concave function $\psi$ yielding $h_j(A) = \psi(m_j(A))$ — performing this for all $j$ yields a $d$-dimensional vector of submodular functions. More generally, any aggregation function $\oplus$ that preserves the submodularity of $h_j(A) = \psi(\oplus_{a \in A} m_j(a))$ would suffice so long as it is also permutation invariant [110] (see Def. 7). For example, with $\oplus = \max$ (i.e., max pooling), the function $\psi(\max_{a \in A} m_j(a))$ is again submodular. We show that there is an expressive infinitely large family of submodular preserving permutation-invariant aggregation operators, taking the form of the weighted matroid rank functions (Def. 6) which are defined based on a matroid $\mathcal{M} = (V, \mathcal{I})$ and a non-negative vector $m \in \mathbb{R}_+^{|V|}$.

**Lemma 1** (Permutation Invariance of Weighted Matroid Rank)**.** *The weighted matroid rank function $rank_{\mathcal{M},m}(\cdot)$ for matroid $\mathcal{M} = (V, \mathcal{I})$ with any non-negative vector $m \in \mathbb{R}_+^{|V|}$ is permutation invariant.*

A proof is in Appendix B. We identify the aggregator $\oplus$ as $\oplus_{a \in A} m_j(a) = rank_{\mathcal{M},m_j}(A)$ and produce new submodular functions $\forall j, h_j^{w_\mathfrak{p}}(A) = \psi(rank_{\mathcal{M},\phi_{w_\mathfrak{p}}(x_:)_j}(A))$, where the $n$-dimensional vector $\phi_{w_\mathfrak{p}}(x_:)_j$ constitute the weights for the matroid. Depending on the matroid, this yields summation (using a uniform matroid), max-pooling (using a partition matroid), and an unbounded number of others thanks to the flexibility of matroids [77]. In theory, this means that the aggregator could also be trained via discrete optimization over the space of discrete matroid structures (in §5, however, our aggregators are fixed). The second stage of the DSPN thus aggregates the modular outputs of the first stage into a resulting $d$-dimensional vector for any set $A \subseteq V$, namely $h_{w_\mathfrak{p}}(A) = (h_1^{w_\mathfrak{p}}(A), h_2^{w_\mathfrak{p}}(A), \ldots, h_d^{w_\mathfrak{p}}(A)) \in \mathbb{R}_+^d$. This can also be viewed as a vector of submodular functions. We presume in this work that each aggregator uses the same matroid and each pillar dimension is used with exactly one matroid, but this can be relaxed leading to a $d' > d$ dimensional space — in other words, many different discrete matroid structures can be used at the same time if so desired.

The third and final "roof" stage of a DSPN is a deep submodular function (DSF) [23]. Deep submodular functions are a nested series of vectors of monotone non-decreasing concave functions composed with each other, layer-after-layer, leading to a final single scalar output. Each layer contained in a DSF has non-negative weight values which assures the DSF is submodular. It was shown in [13] that the family of submodular functions expressible by DSFs strictly increase with depth. DSFs are also trainable using gradient-based methods analogous to how any deep neural network can be trained. A DSF is defined as a set function, but we utilize the "root"[2] [13] multivariate multi-layered concave function $\psi_{w_{\mathfrak{r}}} : \mathbb{R}^d \to \mathbb{R}_+$ of a DSF in this work. That DSPNs significantly extend DSFs, as well as a DSF primer, is further described in §C. The final stage of the DSPN composes the output of the second stage with this final DSF via $f_w(A) = \psi_{w_{\mathfrak{r}}}(h_{w_{\mathfrak{p}}}(A))$ where $w = (w_{\mathfrak{p}}, w_{\mathfrak{r}})$ constitute all the learnable continuous parameters of the DSPN (assuming the discrete parameters, such as the matroid structure, is fixed). While the roof parameters $w_{\mathfrak{r}}$ must be positive, the pillar parameters $w_{\mathfrak{p}}$ are free, but the pillar must produce non-negative outputs.

**Theorem 2** (A DSPN is monotone non-decreasing submodular). *Any DSPN $f_w(A) = \psi_{w_{\mathfrak{r}}}(h_{w_{\mathfrak{p}}}(A))$ so defined is guaranteed to be monotone non-decreasing submodular for all valid values of $w$.*

The proof of Theorem 2 is found in Appendix §D. We immediately have the following corollary.

**Corollary 3** (Submodular Preservation of Weighted Matroid Rank Aggregators). *Any weighted matroid rank function for matroid $\mathcal{M} = (V, \mathcal{I})$ with any non-negative vector $m \in \mathbb{R}_+^{|V|}$, when used as an aggregator in a DSPN, will be submodularity preserving.*

Clearly there is a strong relationship between DSPNs and Deep Sets [110]. In fact any DSPNs is a special case of a Deep Set, but there is an important distinction, namely that a DSPN is assuredly submodular. It may seem like this would restrict the family of aggregation functions available but, as shown above, there are in theory an unbounded number of aggregators to study. Training a DSPN also preserves submodularity simply by ensuring $w_{\mathfrak{r}}$ remains positive which can be achieved using projected gradient descent (the positive orthant is one of the easiest of the constraints to maintain). A deep set, when trained, can easily lose the submodular property, something we demonstrate in §5, rendering the deep set inapplicable when submodularity is required. Also, as was shown in [13], DSFs are already quite expressive, the only known limitation being their inability to express certain matroid rank functions. With a DSPN, however, the aggregator functions immediately eliminate this inability.

**Corollary 4** (DSPN Family). *Let $\mathcal{F}_{DSPN}$ (resp. $\mathcal{F}_{DSF}$) be the family of DSPN (resp. DSF) submodular functions. Then $\mathcal{F}_{DSPN} \supset \mathcal{F}_{DSF}$.*

It is an open theoretical problem if a DSPN, by varying in parameter $w$ and combinatorial space $\mathcal{M}$, can represent all possible monotone non-decreasing submodular functions.

## 3   Peripteral Loss

We next describe a new "peripteral" loss to train a DSPN and simultaneously address how to learn from numerically graded pairwise comparisons (GPCs) queryable from a target oracle.

We are given two objects $E$ and $M$ and we assume access to an oracle that when queried provides a numerical score $Score(E, M) \in \mathbb{R}$. As shorthand, we define $\Delta(E|M) = Score(E, M)$. We do not presume it is always possible to optimize over $\Delta(E|M)$. This score, as mentioned above, could come from human annotators giving graded preferences for $E$ vs. $M$, or could be an expensive process of training a machine learning system separately on data subsets $E$ and $M$ and returning the difference in accuracy on a development set. In the context of DSPN learning, our score comes from the difference of a computationally challenging target submodular function $f_{\mathfrak{t}}$. As an example, $f_{\mathfrak{t}}$ could be a facility location function that requires $O(n^2)$ time and memory and is not streaming friendly, where we have $\Delta(E|M) = f_{\mathfrak{t}}(E) - f_{\mathfrak{t}}(M)$, this being a positive if $E$ is more diverse than $M$ and negative otherwise. Given a collection of $E, M$ pairs (§4), we wish to efficiently transfer knowledge from the oracle target's graded preferences $\Delta(E|M)$ to the learners graded preferences $\delta_{f_w}(E|M) = f_w(E) - f_w(M)$ by adjusting the parameters $w$ of the learner $f_w$.

We are inspired by simple binary linear SVMs that learn a binary label $y \in \{+1, -1\}$ using a linear model $\langle \theta, x \rangle$ for input $x$ and a hinge loss $hinge(z) = \max(1 - z, 0)$. We express the loss of a given

---

[2]"root" here is distinct from "roof".

prediction for $x$ as $hinge(y\langle\theta, x\rangle)$ where $y\langle\theta, x\rangle$ is positive for a correct prediction and negative otherwise, regardless of the sign of $y$.

Addressing how to measure $\delta_{f_w}(E|M)$'s divergence from $\Delta(E|M)$, multiplying the two (as above) would have the same sign benefit as the linear hinge SVM — a positive product indicates the preference for $E$ vs. $M$ is aligned between the oracle and learner. We want more than this, however, since we wish to learn from GPC. Instead, therefore, we measure mismatch via the ratio $\delta_{f_w}(E|M)/\Delta(E|M)$. Immediately, we again have a positive quantity if preferences between oracle and target are aligned, but here grading also matters. A good match makes $\delta_{f_w}(E|M)/\Delta(E|M)$ not only positive but also large. If $\Delta(E|M)$ is large and positive, $\delta_{f_w}(E|M)$ must also be large and positive to make the ratio large and positive, while if $\Delta(E|M)$ is small and positive, $\delta_{f_w}(E|M)$ must also be positive but need not be too large to make the ratio large. A similar property holds when $\Delta(E|M)$ is negative. That is, if $\Delta(E|M)$ is large and negative, then so would $\delta_{f_w}(E|M)$ need to be to make the ratio large and positive, while $\Delta(E|M)$ being small and negative allows for $\delta_{f_w}(E|M)$ to be small and negative while ensuring the ratio is large and positive.

On the other hand, a ratio alone is not a good objective to maximize, one big reason being that small changes in the denominator can have disproportionately large effects on the value of the ratio, which can amplify noise and lead to erratic optimization behavior.[3] Ideally, we could find a hinge-like loss that will penalize small ratios and reward large ratios. Consider the following:

$$\mathcal{L}_\Delta(\delta) = |\Delta| \log(1 + \exp(1 - \frac{\delta}{\Delta})) \tag{1}$$

Here, if $\Delta$ is (very) positive, then $\delta$ must also be (very) positive to produce a small loss. If $\Delta$ is (very) negative then $\delta$ also must be small to produce a small loss. The absolute value of $\Delta$ determines how non-smooth the ratio is so we include an outer pre-multiplication by $|\Delta|$ to ensure the slope of the loss, as a function of $\delta$, asymptotes to negative one (or one when $\Delta$ is negative) in the penalty region, analogous to hinge loss, and thus produces numerically stable gradients. Also analogous to hinge, we add an extra "1" within the $\exp()$ function as a form of margin confidence, analogous to hinge loss that, as with SVMs, expresses a distance between the decision boundary and the support vectors. Plots of $\mathcal{L}_\Delta(\delta)$ are shown in Fig. 7 in Appendix F — this appendix also discusses the smoothness of $\mathcal{L}_\Delta(\delta)$ via its gradients, describes a few additional useful hyperparameters, and provides a numerically stable solution to the case when $\Delta = 0$ (thereby solving any $\delta/0$ issues).

Comparing with standard contrastive losses and deep metric learning [8], a peripteral loss on $\delta_{f_w}(E|M) = f_w(E) - f_w(M)$ can be seen as a high-order self-normalizing by $\Delta$ generalization of triplet loss [108] if $E$ is a heterogeneous and $M$ a homogeneous set of objects. We recover aspects of a triplet loss if we set $E = \{v, v^-\}$ and $M = \{v, v^+\}$ where $M$ is the positive pair (close and homogeneous) while $E$ is the negative pair (distant from each other and thus a diverse pair). With $|E| > 2$ and $|M| > 2$, we naturally represent high order relationships amongst **both** the positive group $M$ and the disperse group $E$ (which is apparently rare, see§A.1). We remind the reader that while our peripteral loss could be used for GPC-style contrastive training for representation learning or for RLHF-based transformer alignment, our experiments in this paper in §5 focus on the submodular learning problem.

### 3.1 Augmented Loss for Augmented Data

Data augmentation is frequently used to improve the generalization of neural networks. Since augmented images represent the same knowledge in principle, they should have the same submodular valuation and also should be considered redundant by any learnt submodular function. Given a set $E$ of images, let $E'$ represent a same-size set of augmented images generated after augmenting each image $e \in E$. This means any learnt model $f_w$ should have $f_w(E) = f_w(E')$ and $\forall e \in E$, $f_w(e) = f_w(e')$. To ensure this is the case, we use an augmentation regularizer $\mathcal{L}_{\text{aug}}(f_w; E, E')$ defined as follows:

$$\mathcal{L}_{\text{aug}}(f_w; E, E') = \lambda_1 (f_w(E) - f_w(E'))^2 + \frac{\lambda_2}{|E|} \sum_{e \in E} (f_w(e) - f_w(e'))^2 \tag{2}$$

Since augmented images represent the same information, the submodular gain of augmented images relative to non-augmented images should be small or zero meaning $f_w(E|E') = f_w(E'|E) = 0$

---

[3]In the sequel, we use $\Delta$ and $\delta$ where the $E, M$ pair is implicit.

and $\forall e \in E$, $f_w(e|e') = f_w(e'|e) = 0$. To encourage this, we use a redundancy regularizer $\mathcal{L}_{\text{redn}}(f_w; E, E')$ defined as follows:

$$\mathcal{L}_{\text{redn}}(f_w; E, E') = \lambda_3(f_w(E|E')^2 + f_w(E'|E)^2) + \tfrac{\lambda_4}{|E|}\sum_{e \in E}(f_w(e|e')^2 + f_w(e'|e)^2) \quad (3)$$

In the above, we square all quantities to provide an extra penalty to larger values, while as quantities such as $f_w(E|E')$ approach zero, we diminish the penalty.

## 3.2 Final Loss

With all of the above individual loss components now well described, our final total loss $\mathcal{L}_{\text{tot}}(f_w; \Delta(E|M), \delta_{f_w}(E|M))$ for an $E, M$ pair becomes:

$$\mathcal{L}_{\text{tot}}(f_w; \Delta(E|M), \delta_{f_w}(E|M)) = \mathcal{L}_{\Delta(E|M)}(\delta_{f_w}(E|M)) + \mathcal{L}_{\text{aug}}(f_w; E \cup M, E' \cup M')$$
$$+ \mathcal{L}_{\text{redn}}(f_w; E, E') + \mathcal{L}_{\text{redn}}(f_w; M, M') \quad (4)$$

With a dataset $\mathcal{D} = \{(E_i, M_i)\}_{i=1}^N$ of pairs, the total risk is:

$$\hat{\mathcal{R}}(w; \mathcal{D}) = \frac{1}{N}\sum_{i \in [N]} \mathcal{L}_{\text{tot}}(f_w; \Delta(E_i|M_i), \delta_{f_w}(E_i|M_i)). \quad (5)$$

Fig. 1 shows the structure of a DSPN and its learning control flow. Please see appendix G for more details.

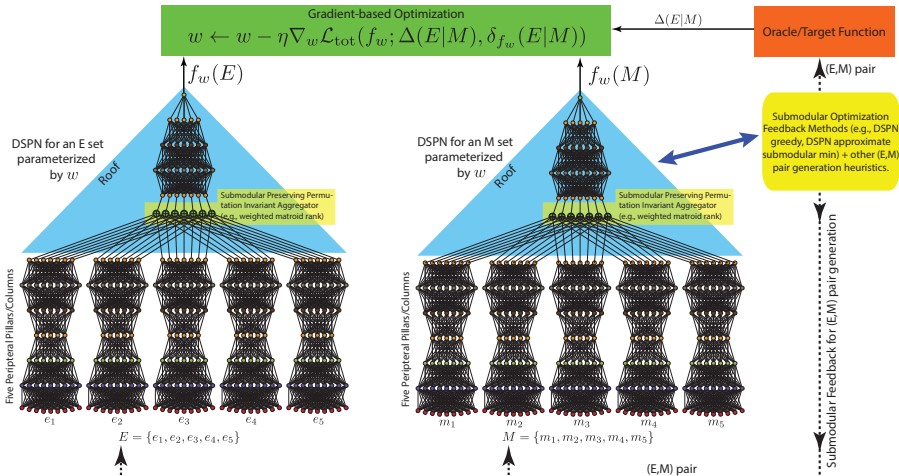

Figure 1: The structure of a DSPN and the control flow of how a DSPN is trained; parameters are shared between the DSPNs processing E and M sets.

# 4 Sampling (E,M) Pairs

DSPN training via the peripteral loss requires a dataset $\mathcal{D} = \{(E_i, M_i)\}_{i=1}^N$ of pairs of sets where each $E_i, M_i \subseteq V$. Since the DSPN starts with the pillar stage, the learnt model can then be tested on samples that are outside of $V$ as we do in §5. Still, exhaustively training on all pairs of subsets from $V$ is infeasible. Therefore, the pairing strategy used to incorporate the sampled sets is a critical component in our dataset construction. We thus propose several practical, both passive and active, strategies to efficiently sample from this combinatorial space. The passive strategies are heuristic-based, while the active strategies are dependent on either the DSPN as it is being learnt or optionally the target function if it is possible to optimize (fortunately, Table 1 shows target sampling is not necessary). Our description here of these strategies is augmented with further discussion and analysis in appendix E.

**Passive Sampling:** If we assume the maximum set size to be $K$ so that $|E| \leq K$ and $|M| \leq K$ and a training dataset to have $C$ classes or clusters, we may generate $E, M$ pairs in two ways. In **Style-I**, we sample a homogeneous set of size $K$ from a randomly chosen single class and a heterogeneous

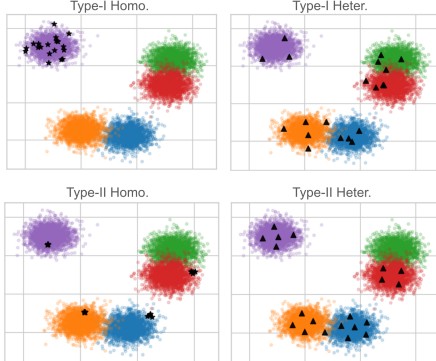

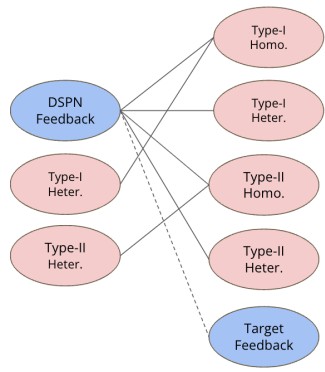

Figure 2: *Passive Sets.* We consider a simple 2D ground set with 5 clusters/classes, as indicated by the colors. The various types of passively sampled sets are depicted (discussed in section 4). Type-I homogeneous sets are randomly sampled from a single class, while Type-I heterogeneous sets are sampled from the full ground set. Meanwhile, Type-II restricts the ground set to a subset of classes and samples "clumps" from each of the sampled classes to construct the homogeneous set, and diverse sets from each class to create the heterogeneous sets. Intuitively, using Type-I/II allows the learnt DSPN to model intraclass/interclass respectively.

Figure 3: *Set Pairing.* We actively sample sets by optimizing the DSPN as it is being learnt or optionally the target function. The depicted graph demonstrates how the actively sampled sets are integrated into $\mathcal{D} = \{(E_i, M_i)\}_i^N$. Th red/blue vertices refer to passively/actively sampled sets. An edge between two vertices indicates that sets from the categories denoted by the vertices are used to create $(E, M)$ pairs to train the DSPN. Dashed edges represent non-critical pairs.

set of size $K$ randomly over the full dataset. In **Style-II**, we first choose a random set of $C' \leq C$ classes, where $C'$ is chosen uniformly at random over $[C]$, and (if $C' < C$) subselect from the ground set a set $V_{C'} \subseteq V$ containing only those samples corresponding to those $C'$ classes. To choose an $E$ set, we perform K-Means clustering on $V_{C'}$, and pick the sample closest to each cluster mean to construct our set. To build an $M$ set, we choose one sample for each class/cluster in $V_{C'}$ and then add its $\lfloor K/C' \rfloor - 1$ nearest neighbors. Importantly, the $E$ and $M$ sets in each pair sampled with **Style-II** are much more difficult to distinguish than **Style-I**. We also note that these sampling strategies are imperfect, so the supposedly heterogeneous $E$ set may not truly be heterogeneous in relation to its homogeneous counterpart $M$. However, as discussed in §E, the peripteral loss allows us to train a DSPN even if the set pairs are imperfectly labeled since the oracle just flips the sign of its preference. Simplified passive 2D sets are shown in Fig. 2 and a set-pairing illustration is seen in Fig. 3.

**Active Sampling:** Sets selected by the passive strategies are independent of both the learner and the target function. Thus, due to the combinatorial nature of the problem, there may be a discrepancy between what sets are valued highly by the DSPN and/or the target. This leads us to employ *actively sampled sets* obtained through two approaches. We refer to the first approach as **DSPN Feedback**, which obtains sets by maximizing or minimizing the DSPN as it is being learnt. As DSPN is submodular, we may optimize it over either full ground set $V$ or a class-restricted ground set to get high-valued subsets, where the greedy algorithm can be used for maximization [73] and submodular minimization [28] can be used to find the homogeneous sets (although in practice, submodular minimization heuristics suffice). We refer to the second approach as **Target Feedback**, where we obtain sets by optimizing the target. The latter approach is only applicable when the target is optimizable (e.g., submodular), which would not be the case with a human oracle. While we do test "Target Feedback" below, our experiments in §5.3 show that it is not necessary for good performance, which portends well for our framework to be used in other GPC contexts (e.g., RLHF learning via human feedback). Thus, while the $E, M$ pair generation strategies mentioned here are specific to DSPN learning, our peripteral loss could be used to train any deep set or other neural network given access to graded oracle preferences, thus yielding a general purpose GPC-style learning procedure.

## 5   Experiments

We evaluate the effectiveness of our framework by training a DSPN to emulate a costly target oracle FL function. We assert that DSPN training is deemed successful if the subsets recovered by maximizing the learnt DSPN are **(1)** assigned high values by the target function (§5.1); and **(2)** are usable for a real downstream task such as training a predictive model (§5.2).

**Datasets:** We consider four image classification datasets: Imagenette, Imagewoof, CIFAR100, and Imagenet100. Imagenette, Imagewoof, and Imagenet100 are subsets of Imagenet-1k [84] where a subset of the original classes is used. Imagenet100 (resp. CIFAR100) have 100 classes and 130,000 (resp. 60,000) images, while Imagenette and Imagewoof have 10 classes (and 13,000 images).

**DSPN Architecture:** We use CLIP ViT encoder [82] for all datasets to get the embeddings that are used as inputs to the DSPN which can be seen as a fixed part of the pillar. We use an additional 2-layer network as the trainable part of the pillar ($\phi_{w_\mathfrak{p}}$); for DSPN roof ($\psi_{w_\mathfrak{r}}$) we use a 2-layered DSF for Imagenette/Imagewoof and 1-layered DSF for CIFAR100/Imagenet100. We use multiple concave activation functions, i.e., all of $\sigma(x) \in \{x, x^{0.5}, \ln(1+x), \tanh(x), 1-e^{-x}\}$ in every layer of the roof (ensuring $w_\mathfrak{r} \geq 0$ during training) and thus allow the training procedure to decide between them. See §H for hyperparameter details.

**Target:** We use Facility Location (FL) as a target whose construction is rather expensive (quadratic) and also does not generalize to held-out data, but this is a key point. That is, we show how we distill from this expensive target FL to a cheaper and generalizable DSPN. Each target FL is created by computing a real-valued feature vector for each sample and then computing a non-negative similarity between each such vector. We used a CLIP ViT encoder to encode input images, and a tuned RBF kernel to construct similarities for our target FLs.

**Training and Evaluation:** We train up to sets of size $|E|, |M| \leq 100$ for all datasets. We show that we generalize beyond this size not just on held-out subsets but even on held-out data. Given ground set $V$, we construct pairs $(E, M)$ both passively and actively, as discussed earlier. Please refer to §G for the details on training. For the target oracle, we use a facility location function with similarity instantiated using a radial basis function kernel with a tuned kernel width. During the evaluation, both learned set functions and target oracle use a held-out ground set $V'$ of images, that is, $V' \cap V = \emptyset$.

## 5.1 Transfer from Target to Learner

We demonstrate that the target FL assigns high values to summaries that are generated by the DSPN in Figure 4. Specifically, we determine if $S$ is a good quality set if it has a high normalized FL evaluation $\frac{f_\mathfrak{t}(S)}{f_\mathfrak{t}(S_{FL})}$, where $f_\mathfrak{t}$ is the target FL and $S_\mathfrak{t}$ is the set generated by applying cardinality constrained greedy maximization to the target FL.

Therefore, we compare different DSPN models based on the normalized FL evaluation that their maximizers attain. We compare DSPN models trained with three different losses: the peripteral loss, the contrastive max-margin style structured prediction loss [98], and a regression loss (shown as *DSPN (peripteral)*, *DSPN (margin)* and *DSPN (regression)* respectively in Figure 4). Please refer appendix H for the definition of baseline losses (Equations 30 and 31). We also compare against randomly generated sets, which are denoted as *Random*. From this set of experiments, we find that the DSPN trained with the peripteral loss consistently outperforms all other baselines across differ-

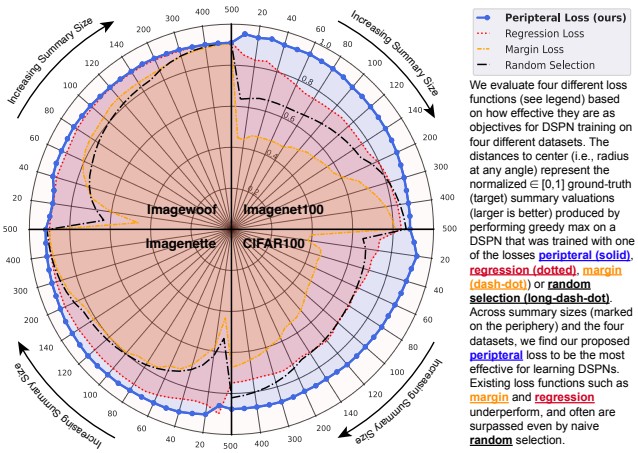

We evaluate four different loss functions (see legend) based on how effective they are as objectives for DSPN training on four different datasets. The distances to center (i.e., radius at any angle) represent the normalized $\in [0,1]$ ground-truth (target) summary valuations (larger is better) produced by performing greedy max on a DSPN that was trained with one of the losses **peripteral (solid)**, **regression (dotted)**, **margin (dash-dot)** or **random selection (long-dash-dot)**. Across summary sizes (marked on the periphery) and the four datasets, we find our proposed **peripteral** loss to be the most effective for learning DSPNs. Existing loss functions such as **margin** and **regression** underperform, and often are surpassed even by naive **random** selection.

Figure 4: *Transfer.* We compare different loss functions in terms of their effectiveness at training a DSPN.

ent **summary sizes** (budgets) and datasets. Interestingly, the DSPN outperforms random even on budgets greater than 100, despite never having encountered such subsets during training. This provides evidence that our GPC-based peripteral loss is the loss of choice for training DSPNs.

## 5.2 Application to Experimental Design

*Experimental design* involves selecting a subset of unlabeled examples from a large pool for labeling to create a training set [80]. In this section, we evaluate the performance of a learnt DSPN based on how well a simple linear model generalizes when trained on a training set that was chosen by a DSPN.

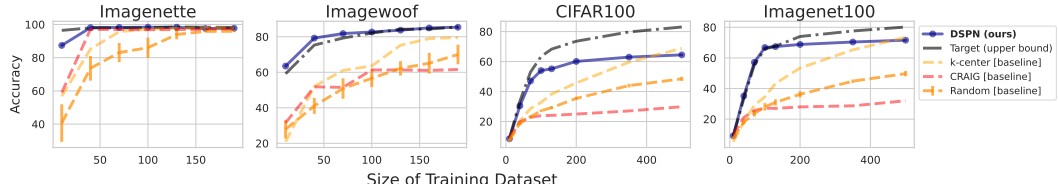

Figure 5: ***Offline Experimental Design.*** We compare different summarization procedures, and assess them based on the test accuracy that a linear model attains upon being trained on the summary. We find that maximizing the learnt DSPN generates high quality datasets, outperforming existing summarization techniques such as CRAIG and k-centers. In many cases, the DSPN approaches the Target FL even though the latter is aware of the duplicates present in the data. **Takeaway: DSPN effectively chooses training samples for labeling from an unlabeled pool.**

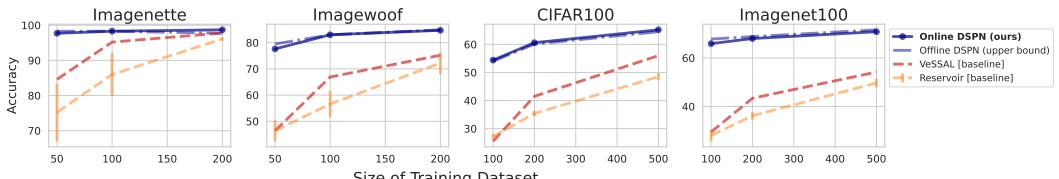

Figure 6: ***Online Experimental Design.*** The performance of a linear probe, trained based on a subset selected by an algorithm that does not use labels, is reported at varying budgets. *DSPN-online* achieves far higher performance than other streaming algorithms such as *reservoir sampling* and *VeSSAL* and is performs comparably to offline algorithms. **Takeaway: DSPN can effectively select training samples for labeling from a *stream* of unlabeled data.**

We consider an *offline setting* where we assume the full unlabeled pool is available and an *online setting* where the unlabeled pool is presented to the model in a non-iid stream. To test the robustness of our framework to real-world challenges, we add duplicates such that the ground set is heavily class-imbalanced (the procedure to generate the class-imbalanced set is shown in Appendix H). Importantly, in these experiments we train with one set $V$ and test on a completely held-out set $V'$ with $V \cap V' = \emptyset$; that is, the testing DSPN is on an entirely different ground set than the training DSPN.

**Offline Setting:** The offline setting assumes that the full unlabeled pool is accessible by the summarization procedure. The results of these experiments are presented in Figure 5. Our approach, denoted as *DSPN* in the figure, applies greedy maximization to the learnt DSPN to generate the summary. We also consider a *cheating* experiment by applying greedy to the target FL after removing duplicates from the ground set; we refer to this approach as *Target FL*. Finally, we consider three baseline summarization procedures CRAIG [69], K-centers [89], and random subset selection [75]. The learnt DSPN achieves performance comparable to the target FL in all datasets up to a budget of 100. Beyond that, the difference between DSPN and the Target FL grows but the DSPN continues to generate higher quality than any of the baselines, showing DSPNs generalize to set sizes beyond the training set sizes.

**Online Setting:** We consider an online setting where the data is presented to the model in a non-iid, class-incremental stream, as done in the continual learning literature [63, 18, 2, 113]. Unlike the facility location function which requires the full ground set for evaluation, the DSPN can be maximized in a streaming manner. Thus, we employ the streaming maximization algorithm proposed by [52] to maximize the DSPN, due to its effectiveness in real-world settings and its lack of additional memory requirements. This approach is referred to as *DSPN-online* in Figure 6.

We compare our approach against two streaming summarization baselines. The first approach, *reservoir sampling* [104], is an algorithm that enables random sampling of a fixed number of items from a data stream as if the sample was drawn from the entire population at once. The second approach, *VeSSAL* [86], seeks to select samples such that the summary maximizes the determinant of the gram matrix constructed from the model embeddings. We also apply (offline) greedy maximization to the target facility location (*Target FL-offline*) and the learned DSPN (*DSPN-offline*), which serve as upper bounds to what is attainable by streaming DSPN maximization.

In Figure 6, we present quantitative results where a linear model is trained on a given streaming summary after labels have been obtained. We find that *DSPN-online* significantly outperforms the other streaming baselines on **all** datasets. Surprisingly, *DSPN-offline* and *DSPN-online* achieve similar

performance, demonstrating the flexibility of DSPNs. *FL-offline* tends to generate the highest quality summaries, but requires access to the full ground set and assumes duplicates are manually removed.

## 5.3 Ablations

Table 1, provides Imagenet100 results on sets of size 100 that explore the importance of various components of the DSPN. We report three numbers for each ablation: (1) *FL Eval* which corresponds to the normalized FL evaluation discussed in § 5.1 (2) *Offline Acc* which denotes the final accuracy of a linear model in the offline experimental design setting and (3) *Online Acc* denotes the final accuracy of a linear model in the online experimental design setting.

| Method | FL Eval | Offline Acc | Online Acc |
|---|---|---|---|
| DSPN | 0.92 | **67.8** | 65.9 |
| w/o target feedback | **0.93** | 67.7 | **66.9** |
| w/o feedback | 0.83 | 47.7 | 30.1 |
| w/o type-II sampling | 0.64 | 13.1 | 10.7 |
| PC rather than GPC | 0.90 | 62.8 | 63.4 |
| w/o learnt pillar | 0.71 | 39.1 | 38.0 |
| Deep Set | 0.56 | 9.2 | 3.5 |
| Set Transformer | 0.34 | 13.8 | – |

Table 1: ***Ablations on Imagenet100***. The importance of feedback, learnt features, enforcing submodularity, and GPC (in each column, **bold** is best, underline is 2nd best). "w/o feedback" indicates a DSPN model trained without feedback, while "w/o target feedback" is when the oracle is only queried. For Set Transformer, streaming maximization fails to produce a summary of the required size (therefore it is omitted). "PC rather than GPC" uses non-graded pairwise comparisons, showing the benefits of grading, consistent with [61]. All results are for a size-100 set.

**Removing (FL) Feedback:** Active selection is obtained by maximizing the DSPN or Target FL during training (denoted as *DSPN feedback* and *Target feedback* respectively). We find that only removing the sets obtained from target feedback has little to no impact on the final performance of the DSPN, suggesting that it is not necessary to optimize the target function. *This shows that our framework can be applied to GPC learning even when it is not possible to optimize the target, such as with human oracles.* However, we find that completely removing Active selection significantly hurts the final DSPN .

**Removing Learnable Pillar:** We assess the utility of a learnable pillar, by training a DSPN with a fully frozen pillar. In this case, we attempt to learn the weights of a DSF on random projections of CLIP features. We discover that the DSPN without learnable pillars significantly underperforms its counterpart. We discuss this further in appendix I .

**Removing Submodularity:** We study the utility of submodular constraints to learn the target. To this end, we compare DSPN against a Deep Set [110] and SetTransformer [55] architecture in the third section of table 1. we discover that the absence of submodularity **significantly hampers** the performance for all the metrics, since the greedy algorithm has no guarantees on an arbitrary set function.

## 6 Conclusion and Future Work

We have proposed DSPNs and methods for training/distilling them. Since the computational cost for querying a DSPN is independent of dataset size, and the training scheme leverages automatically generated $(E, M)$ pairs, our methods learn practical submodular functions scalably. It may ultimately be possible to scale our training framework to massive heterogeneous datasets to develop foundation DSPNs having strong *zero-shot summarization* capabilities on a wide array of domains. We also propose the *peripteral loss* which leverages numerically graded relationships between pairs of objects. The graded pairwise preferences are provided from an oracle and are used to learn a submodular function that captures these relationships. However, the scope of the peripteral loss extends far beyond training DSPNs and has many applications such as new contrastive representation learning procedures, and learning a reward model in the context of RLHF with real human oracles.

## 7 Acknowledgments

We thank Suraj Kothawade, Krishnateja Killamsetty and Rishabh Iyer for several early discussions. This work is supported in part by the CONIX Research Center, one of six centers in JUMP, a Semiconductor Research Corporation (SRC) program sponsored by DARPA and by NSF Grant Nos. IIS-2106937 and IIS-2148367.

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

# A   Other Related Work

In addition to the previously discussed literature in §1 that addresses learning through pairwise comparison, human preference elicitation, and contrastive learning, this section provides a comprehensive review of both theoretical and empirical studies related to the learning of set functions.

## A.1   Relationship to contrastive learning

We are inspired by contrastive learning [8], which has pushed the boundaries of deep learning by leveraging the massive information in unlabeled data. Unlike standard supervised learning which teaches models to associate samples to human-supplied labels, contrastive learning encourages models to capture relationships between pairs, groups, or sections of unlabeled samples. It does this by "contrasting" positive pairs (i.e., similar data points) against negative pairs (dissimilar data points). Mathematically, this approach uses loss functions that encourage the model to learn embeddings where positive pairs become "close" (according to some distance measure), while negative pairs become "far." A common setting is the contrastive loss, such as the triplet loss or the Siamese network loss. For instance, in the triplet loss framework, the model is trained on triplets of data points consisting of an anchor $x$, a positive example $x^+$ (similar to the anchor), and a negative example $x^-$ (dissimilar from the anchor). Optimizing a model with such a loss function encourages a representation that respects these positive/negative pairwise preferences.

On the other hand, such contrastive approaches are analogous to binary-only pairwise comparison elicitations. That is, existing contrastive losses neither utilize information regarding how positive the pair $x, x^+$ is nor how negative the pair $x, x^-$ is. A more nuanced graded comparison, if available, would be an improvement. In fact, the benefit of graded comparisons is analogous to the benefits of using soft over hard labels in supervised training (say from model distillation) which has benefits in uncertainty representation, regularization, noise mitigation, adjustable smoothness via softmax temperature, and more efficient knowledge transfer. Another limitation of previous contrastive learning is that it operates only with positive or negative groups of size two (pairs) rather than larger groups. While it is often the case that the anchor $x$ is contrasted with many negative examples, it is always done so pairwise, considering each $x^-$ with $x$ together in a sum of pairwise terms. Also, while [100, 42] is the only work we are aware of that considers more than two positive examples at a time, each of these positive samples is again considered only paired up with a negative sample, leading to $\binom{k}{2}$ pairings with $k$ positive examples. Sums of pairwise terms do not allow for the consideration of higher order relationships, e.g., a given pair might be measured differently in the context of a third sample. In addition to addressing the two aforementioned problems, DSPNs and our GPC-style loss addresses this issue as well, although our learning goal in this paper is learning submodularity not in representation learning (the latter of which we leave to future work).

## A.2   Learning Non-submodular Set functions

Recently, there has been a growing focus on leveraging data-driven neural methods for estimating set functions, with practical applications spanning anomaly detection, amortized clustering of mixture models, and point cloud classification [81]. Common to these applications is the fundamental approach of embedding each element within a set into dense embeddings, followed by permutation-invariant aggregation (set-to-vector transformation), and subsequent integration into a deep neural network [110, 38, 83]. The significance of incorporating interaction among set elements has been emphasized by [55], who argue that conventional aggregations, such as average-pooling, can be interpreted as a specialized form of attention. Consequently, they propose an attention-based architecture known as the set-transformer. Building upon this, [67] have made computational enhancements to the attention-based architecture. Lastly, there has been a recent exploration into a probabilistic interpretation of set-to-vector transformations [113, 46], further refining the methodology.

The works mentioned above diverge from our approach for several reasons. Firstly, they primarily focus on supervised learning oracles utilizing static datasets to learn neural parameters, whereas our method draws inspiration from max-margin contrastive learning and employs a dynamically updated dataset. Secondly, these existing works lack architectural and/or optimization constraints ensuring the submodularity of the learned set function, in contrast to our work on learning deep submodular functions, where submodularity is explicitly guaranteed. However, for a comprehensive evaluation against existing literature, we include Deep Sets and Set-Transformer as a baseline for comparison. Thirdly, the set sizes used in these works are much smaller (30 in many experiments)

compared to ours which go up to 500. Lastly, the learned set function in these works lacks a clear interpretation associated with extremal—sets that (locally) maximize or minimize the set function. In contrast, our Deep Submodular Function (DSF) yields heterogeneous/homogeneous sets upon maximization/minimization. It is noteworthy that the choice of a submodular function as the inductive bias may not be essential for certain tasks, such as point-cloud classification and amortized clustering of mixture models. However, it proves to be a plausible inductive bias in scenarios where the objective is to count unique characters [110, 113, 46, 55], since it can be exactly be expressed as a partition matroid rank function, which lies in the family of deep submodular function.

Beyond neural approaches, one of the previous works [66] has also considered learning Determinantal Point Process, or DPPs [50], by learning the $\mathcal{O}(n^2)$ similarity matrix by noise contrastive estimation [35]. However, their learning procedure is different from ours, and sampling from a DPP is much more computationally expensive than maximizing a submodular function.

### A.3 Theoretical Work on Learning Submodular Functions

Numerous works have investigated strategies for understanding submodular functions from a theoretical perspective. [31] examined the approximation of an oracle for every subset, utilizing a multiplicative approximation approach to approximate the submodular polyhedron of the oracle using Löwner ellipsoids. In contrast, [5, 6] focused on approximating the oracle with high probability on sets sampled from specific distributions (often referred to as Probably Mostly Approximately Correct (PMAC) learning) by transforming the problem into the identification of linear separators. Notably, [6] demonstrated the impracticality of approximating any arbitrary monotone non-decreasing submodular function to arbitrary closeness under PMAC learning in polynomial queries. Subsequently, [7] proposed an algorithm for learning the oracle through pairwise comparisons in polynomial queries. While this work shares theoretical commonalities with ours, it is important to highlight that [7] examined set pairs sampled from a static distribution, contrasting with our approach, which involves a dynamically changing distribution of set pairs due to our submodular feedback. Remarkably, across the aforementioned works, the learned functions exhibit a form that can be represented easily using a Deep Submodular Function (DSF), a facet that our work incorporates and explores further.

### A.4 Neural Estimation of Submodular Functions

Since submodular functions have $2^{|V|}$ degrees of freedom, empirical approaches of learning submodular functions restrict the search space to a parametric subclass such as feature-based functions or DSFs. Early data-driven learning techniques use the max-margin structured prediction framework introduced in [98], which requires human annotators to manually construct summaries/sets that should be assigned high values by the learnt submodular function [60, 102, 93, 36, 29]. These approaches have consistently demonstrated that learned coefficients, as opposed to handcrafted ones, result in submodular functions that can generate much higher quality summaries. However, these approaches depend on human-constructed summaries which can be prohibitively expensive to procure at scale. Furthermore, that the features suitable for the feature-based function are *not* learned, unlike our work where we jointly learn features and associated parameters for the submodular function. Despite the empirical success of this framework, these approaches all require sets chosen by humans which are difficult to procure at scale.

More recently, [22] proposed to estimate a submodular function using recursive composition with a learned concave function, arguing that the lack of a "good" concave function is a weakness of DSF (and only used one concave function in their experiments). In our work, we alleviate this by showing that one can learn a good DSF by training with multiple concave functions simultaneously with varying degrees of saturation. It should also be noted that there is no theoretical discussion if the representation capacity of the function class strictly increases with more recursive steps, unlike DSF, where the capacity strictly increases with additional layers. Furthermore, they learn the parameters in a supervised manner, given the annotated high-valued summaries, unlike our work which does so in a self-supervised fashion. Lastly, the learned modular function for DSF has the interpretation of capturing concepts (appendix I) which is not in their case. Remarkably, it is an interesting extension where one combines the technique proposed in [22] to learn the concave function jointly with our technique for learning the parameters.

[51] proposes an unsupervised framework to learn submodular functions, similar to our work. Their approach involves the maximization of a handcrafted objective function, evaluating the quality of

weights through measurements of submodular function properties like curvature and sharpness. Notably, they do not engage in the simultaneous joint learning of features alongside parameters and instead they first train an auto-encoder to get the features. To the best of our knowledge, no existing studies have effectively undertaken the joint learning of submodular function parameters and features, a key aspect of our proposed methodology.

Finally, we note the existence of related work using the same nomenclature, DSPN, introduced by [112], which stands for Deep Set Prediction Network. The primary objective of [112] is to predict a set based on dense features. In contrast, our research focuses on the development of a deep submodular function (DSF) capable of generating sets through optimization.

## B  Submodular Functions and Weighted Matroid Rank

Submodular set functions effectively capture notions of diversity and representativeness. Continuing on from §1, it is known that they are useful in machine learning in the context of data subset selection [69, 45, 44, 106], feature selection [62], summarization [59], active learning [106, 49], experimental design [11], more recently in reinforcement learning [79] and several other areas that involve some form of discrete optimization. Submodular functions also appear naturally in many other areas, such as theoretical computer science and statistical physics [101, 12].

Submodular functions are also of interest since they enjoy properties sufficient for efficient optimization. For a ground set of size $n$, they lie in a cone of dimension $2^n$ in $\mathbb{R}^{2^n}$, but can still be minimized exactly without constraints in polynomial time [28]. Even though submodular function maximization is NP-hard, it offers a fast constant factor approximation algorithm, for instance in the cardinality-constrained case, greedy results in $1 - 1/e$ guarantee for polymatroid functions [73, 68]. Similarly one can give guarantees for other constrained cases such as knapsack constraint or more complicated independence or matroid constraints [15, 16, 54, 40, 39].

In this section, we offer a brief primer on submodularity and their relationship to matroid rank.

Any function $f : 2^V \to \mathbb{R}$ that maps from subsets of some underlying ground set $V$ of size $|V| = n$ is called a discrete set function. For a set function to be *submodular*, it must satisfy the property of diminishing returns. This means that for any $X \subseteq Y \subseteq V$ and any $v \in V \setminus Y$ the gain of adding $v$ to the smaller set is larger than the gain of adding $v$ to the larger set, meaning:

$$f(v|X) \triangleq f(X \cup \{v\}) - f(X) \geq f(Y \cup \{v\}) - f(Y) \triangleq f(v|Y) \tag{6}$$

We use the (conditional) gain notation $f(v|X) \triangleq f(X \cup \{v\}) - f(X)$ regularly in the paper. We also assume also that all submodular functions are monotone non-decreasing (or just monotone), which means that for any $X \subseteq Y$ we have that $f(X) \leq f(Y)$ and non-negative, meaning that $0 \leq f(X)$ for any $X \subseteq V$. We also assume $f$ is normalized so that at the emptyset $f(\emptyset) = 0$. Such normalized monotone non-decreasing submodular functions necessarily are always non-negative, and are commonly referred to as polymatroid functions [12, 20, 21, 64].

If Eqn (6) holds with equality for all sets, then the function is referred to as a *modular* function which implies that $f$ can be equated with a length $n$ vector, that is for all $X \subseteq V$, $f(X) = \sum_{x \in X} f(x)$, which means the vector associated with $f$ is $(f(v) : v \in V) \in \mathbb{R}^n$. We often use simplified notation for modular functions — that is, given a length $n$ vector $m \in \mathbb{R}^n$, we can define the modular function $m : 2^V \to \mathbb{R}$ as $m(A) = \sum_{a \in A} m(a)$.

If the inequality in Eqn (6) holds in the reverse, meaning

$$f(v|X) \triangleq f(X \cup \{v\}) - f(X) \leq f(Y \cup \{v\}) - f(Y) \triangleq f(v|Y) \tag{7}$$

then the function is known as a *supermodular* function.

For completeness, we note that an equivalent definition of submodularity would say that for any two sets $A, B \subseteq V$, we have that $f(A) + f(B) \geq f(A \cup B) + f(A \cap B)$. It is not hard to show that this and Inequality (6) are identical [12, 28].

Given a finite set $V$ and a set of subsets $\mathcal{I} = \{I_1, I_2, \dots\}$, the pair $(V, \mathcal{I})$ is known as a *set system*.

Matroids [53, 77, 88] are useful algebraic structures that are strongly related to submodular functions [28]. Given a finite ground set $V$ of size $|V| = n$, a matroid $\mathcal{M}$ is a set system that consists of

the pair $(V, \mathcal{I})$ where $V$ is said ground set and $\mathcal{I} = \{I_1, I_2, \ldots\}$ is a set of subsets of $V$, meaning $I_i \subseteq V$ for all $i$. To define matroid $\mathcal{M}$ we say $\mathcal{M} = (V, \mathcal{I})$. The subsets $\mathcal{I}$ are known as the independent sets (meaning, for any $I \in \mathcal{I}$, $I$ is said to be independent) and they have certain properties that make this the set system interesting and useful. Formally, a matroid is defined as follows:

**Definition 5** (Matroid). *Given a ground set $V$ of size $|V| = n$, a matroid $\mathcal{M} = (V, \mathcal{I})$ is a set system where the set of subsets $\mathcal{I} = (I_1, I_2, \ldots,)$ have the following properties:*

1. $\emptyset \in \mathcal{I}$

2. *If $I \in \mathcal{I}$ and $I' \subset I$, then $I' \in \mathcal{I}$*

3. *For any $I, I' \in \mathcal{I}$ with $|I| < |I'|$, there exists a $v \in I' \setminus I$ such that $I \cup \{v\} \in \mathcal{I}$.*

Matroids generalize the algebraic properties of vectors in a vector space, where independence of a set corresponds to sets of vectors that are linearly independent. If we think $V$ as a set of indices of vectors, and $A = \{a_1, a_2, \ldots\} \subseteq V$, then the $A$ being independent corresponds to the vectors associated with $A$ being linearly independent. Matroids however capture the algebraic properties of independence and there are valid matroids that can not be representable by any vector space. Note also that the algebraic structure is similar to that of collections of random variables that might or might not be statistically independent.

While there are many types and useful properties of matroids, the rank of a matroid measures the independence of a set based on the largest independent set that it contains, namely

$$\text{rank}_{\mathcal{M}}(A) = \max_{I \in \mathcal{I}} |A \cap I| = \max\{B : B \subseteq A \text{ and } B \in \mathcal{I}\} \tag{8}$$

For any matroid, the rank function is submodular, meaning for any $A, B \subseteq V$, $\text{rank}_{\mathcal{M}}(A) + \text{rank}_{\mathcal{M}}(B) \geq \text{rank}_{\mathcal{M}}(A \cup B) + \text{rank}_{\mathcal{M}}(A \cap B)$. For example, the function $f(A) = \min(|A|, \alpha)$ with integral valued $\alpha$ is a classic concave-composed with modular function (which is thus submodular) but is also the rank function for partition matroid with one partition block and limit of $\alpha$. A matroid rank function is also a polymatroid function since $\text{rank}_{\mathcal{M}}(\emptyset) = 0$ and $\text{rank}_{\mathcal{M}}(A) \leq \text{rank}_{\mathcal{M}}(B)$ whenever $A \subseteq B$.

A matroid rank functions can be significantly extended when it is associated with a non-negative vector $m \in \mathbb{R}_+^{|V|}$ of length $|V| = n$. With such a vector, one can define a weighted matroid rank function that returns the maximum weight of any base of a set $A$:

**Definition 6** (Weighted Matroid Rank Function). *Given a matroid $\mathcal{M} = (V, \mathcal{I})$ and a non-negative vector $m \in \mathbb{R}_+^n$, the weighted matroid rank function for this matroid and vector is defined as:*

$$rank_{\mathcal{M}, m}(A) = \max_{I \in \mathcal{I}} m(A \cap I) \tag{9}$$

A weighted matroid rank function is still submodular and is still a polymatroid function. In fact, weighted matroid rank functions generalize many much more recognizable functions. Here are several examples.

1. Consider a free matroid $M = (V, \mathcal{I})$ where $\mathcal{I} = 2^V$. This means that all $2^n$ subsets of $V$ are independent in the matroid. Then given non-negative vector $m \in \mathbb{R}_+^{|V|}$, the weighted matroid rank function has the form $\text{rank}_{\mathcal{M}, m}(A) = \sum_{a \in A} m(a)$, thus the weighted matroid rank function is just the modular function $m$.

2. Consider a one-partition matroid rank function with limit $k = 1$. This is a matroid $M = (V, \mathcal{I})$ where $\mathcal{I} = \{A \subseteq V : |A| = 1\}$, meaning all sets of size one are independent but no other set is independent in the matroid. Given non-negative vector $m \in \mathbb{R}_+^{|V|}$, the weighted matroid rank function has the form $\text{rank}_{\mathcal{M}, m}(A) = \max_{a \in A} m(a)$.

3. The facility location function can be seen as the sum of a weighted matroid rank functions using the same matroid as the previous example but using different weights for each term in the sum. That is, given a similarity matrix $[S]_{ai}$ of non-negative entries, the facility location function has form $f(A) = \sum_{i=1}^n \max_{a \in A} s_{ai}$. Hence, using the same matroid as the previous case, the facility location takes the form $f(A) = \sum_{i=1}^M \text{rank}_{\mathcal{M}, s_{:,i}}(A)$ where $s_{:,i}$ is the $i^{\text{th}}$ column of the matrix.

Another interesting (and apparently previously unreported) property of weighted matroid rank functions is that they are always permutation invariant in the same sense as a deep set function [110] is permutation invariant. Firstly, we restate the definition of permutation invariance from [110].

**Definition 7** (Permutation Invariance [110]). *A function $f : 2^V \to \mathbb{R}$ acting on sets is said to be permutation invariant if the function is invariant to the order of objects in the set. That is for any set $A \subseteq V$ with $A = \{a_1, a_2, \dots, a_m\}$, and for any permutation $\pi : A \to A$, we have that for any set $f(\{a_1, a_2, \dots, a_m\}) = f(\{\pi(a_1), \pi(a_2), \dots, \pi(a_m)\})$.*

Thus, permutation invariance means that the order in which elements are considered or arranged does not affect the outcome of the function that is applied to a set. For the weighted matroid rank function, this means that if we take a subset $A \subseteq V$ and permute its elements, the $\text{rank}_{\mathcal{M},m}(A)$ stays the same. The reason is because the rank function is concerned only with the elements in the subset, not the order in which they are listed. Thus, permutation invariance aligns with the properties of matroids as we formalize in the next lemma.

**Lemma 1** (Permutation Invariance of Weighted Matroid Rank). *The weighted matroid rank function $\text{rank}_{\mathcal{M},m}(\cdot)$ for matroid $\mathcal{M} = (V, \mathcal{I})$ with any non-negative vector $m \in \mathbb{R}_+^{|V|}$ is permutation invariant.*

*Proof.* The weighted matroid rank function for matroid $\mathcal{M} = (V, \mathcal{I})$ is defined as

$$\text{rank}_m(A) = \max_{I \in \mathcal{I}} m(A \cap I) = \max_{I \in \mathcal{I}} \sum_{a \in A \cap I} m(a) \tag{10}$$

and since nowhere in this definition are the order of the elements in $A$ used, but rather only a sum over some of the elements as shown on the right hand side of the above, the function is permutation invariant. This also immediately follows given Proposition 8. $\square$

In fact, this property is true for any submodular function.

**Proposition 8.** *Submodularity and Permutation Invariance. Any submodular set function $f : 2^V \to \mathbb{R}$ is permutation invariant.*

## C   Primer on and Novelty over Deep Submodular Functions

In this section, we offer a brief background on deep submodular functions (DSFs) and demonstrate how our results in the present paper are **not** just incremental over this previous work — rather the present paper offers significant novelty over, as far as we can tell from an extensive literature search, previously published papers.

[96] proposed a scalable algorithm for minimizing a class of submodular functions known as *decomposable functions*. In general, given a collection of modular functions, $\{m_i\}_{i=1}^N$ where each $m_i : V \to \mathbb{R}_+$ and $\phi$, a monotone non-decreasing, normalized $\phi(0) = 0$ and non-negative concave function,

$$f(A) = \sum_{i=1}^N \phi(m_i(A))$$

is said to be a "decomposable" function.

In machine learning, objects of interest are often represented by featurizing them into a finite-dimensional vector space. If $U = \{u_1, u_2, \dots, u_{|U|}\}$ indexes the dimensions, then for any example $v \in V$ the modular function $m_U(v) = \left(m_{u_1}(v), m_{u_2}(v), \dots, m_{u_{|U|}}(v)\right)^T$ represents the features of $v$. $m_u(v)$ can be interpreted in several ways such as the degree of presence of $u - th$ concept in $v$ similar to concept bottleneck models [48] or even the count of an n-gram feature [107, 59, 60]. With this nomenclature, we re-define *decomposable* submodular functions as feature-based functions that has the form:

$$f(A) = \sum_{u \in U} w_u \phi_u(m_u(A)), \tag{11}$$

where $w$ represents a non-negative weight for each feature. The gain of adding examples with more of the property represented by feature $u$ diminishes as the set gets larger due to concave composition and

the diminishing returns of the concave function. Therefore, intuitively the set $X \subseteq V$ that maximizes the feature-based function would comprise examples that produce a diverse set of features according to the features' properties.

One issue with the feature-based function happens when there is redundancy between two features $u_i$ and $u_j$. For instance, if the features are a count of n-grams, and since higher-order n-grams encompass lower-order n-grams, having too many of a certain high-order n-gram would also mean having many of its lower-order n-grams, leading to redundancy (or correlation) in what the features represent. As eq. (11) suggests, feature-based functions treat each feature independently. This motivates a flexible class of submodular functions, known as *Deep submodular functions* or DSFs [13] which mitigates the problem of redundancy by an additional concave composition followed by weighted aggregation which allows for interaction between sets of redundant features as follows:

$$f(A) = \sum_{s \in S} \omega_s \phi_s \left( \sum_{u \in U} w_{s,u} \phi_u \left( m_u(A) \right) \right) \tag{12}$$

Here, $w_{s,u}$ and $\omega_s$ are non-negative mixture weights. This naturally leads to a representation resembling neural networks when we iteratively apply a concave function followed by linear aggregation, giving rise to Deep Submodular Functions. $\forall s \in S$, [13] define $\sum_{u \in U} w_{s,u} \phi_u \left( m_u(A) \right)$ as meta-features, and in general one can stack more layers of concave composition followed by affine transformations, leading to meta-meta features, and so on. Let $L$ be the total number of layers in a DSF; each meta-feature at layer $\ell$ of this network is a submodular function, we index them using $U^{(\ell)}$, which means in the example in equation (12) $U = U^{(0)}$ and $S = U^{(1)}$.

$W^{(\ell)} \in \mathbb{R}_+^{|U^{(\ell)}| \times |U^{(\ell-1)}|}$ be the mixture weights at layer $\ell$, where for the last layer $L$ we have a uniform weight, which means $W^L \in \mathbb{R}_+^{|U^{(L)}|}$ Concave functions applied at the $j$-th submodular function of layer $\ell$ is represented by $\phi_j^{(\ell)}$. To write concave functions in vector form, we define $\Phi_U : \mathbb{R}^{|U|} \to \mathbb{R}^{|U|}$ such that for any $\mathbf{a} \in \mathbb{R}^{|U|}$ we define $\Phi_U(\mathbf{a})$

$$\Phi_U(\mathbf{a}) \triangleq [\phi_1(\mathbf{a}_1), \phi_2(\mathbf{a}_2), \ldots, \phi_{|U|}(\mathbf{a}_{|U|})]^T \tag{13}$$

With the notation above, for $X \subseteq V$, DSF $f(A)$ is defined as following

$$f(A) = W^{(L)} \cdot \Phi_{U^{(L-1)}} \left( W^{(L-1)} \cdot \Phi_{U^{(L-2)}} \left( \ldots W^{(1)} \cdot \Phi_{U^{(0)}} \left( m_{U^{(0)}}(A) \right) \ldots \right) \right) \tag{14}$$

or recursively as follows

$$f^L(A) = W^{(L)} \cdot \Phi_{U^{(L-1)}} \left( f^{L-1}(A) \right) \tag{15}$$

where,

$$f^1(A) = W^{(1)} \cdot \Phi_{U^{(0)}} \left( m_{U^{(0)}}(A) \right) \tag{16}$$

here, "$\cdot$" denotes matrix multiplication. To maintain submodularity, all weight matrices must be non-negative. A DSF can incorporate skip connections, and in general, a directed acyclic graph topology can be used to represent the DSF architecture with the help of recursive definitions (refer to section 4.1 of [13]).

Lastly, [13] showed that DSFs strictly generalize feature-based functions, and adding additional layers strictly increases the representation capacity, therefore suggesting the utility of additional layers.

It is important to realize that our present work on DSPNs and GPC-based learning is not just an incremental improvement to DSFs, but rather offers a new class of submodular function and an entirely new way to learn the submodular function. There are two pre-existing papers on DSFs, this paper [23] from 2016 as well as a more theoretical paper [13] from 2017. The first paper [23] introduces DSFs and uses a max-margin based learning algorithm that was introduced by [60] in their "Submodular Shells" paper but applies that learning algorithm to DSFs. Max-margin learning of submodular functions requires a dataset of highly valued subsets according to some unknown submodular function to be learnt (e.g., each subset can be thought of as a good summary) and the goal of max-margin learning in this case is to learn a submodular function so that it scores those subsets highly relative to other subsets. Max-margin learning does not care at all about the rest of the valuations of the submodular function (e.g., a max-margin learnt submodular function is under

no obligation to perform well for problems other than submodular maximization, and is under no obligations to preserve the rank valuations of a given teacher submodular function).

Our present paper is entirely different from these two papers. While our DSPN "roof" structure uses the deep concave function from DSFs, there is no commonality beyond this. In the present (DSPN) paper, we have introduced a new parametric submodular function class (DSPNs) that overcomes the only known weakness of DSFs (namely, DSFs inability to express certain graphic matroid rank functions, as proven in [13]). DSPNs do not have this weakness thanks to the submodularity-preserving permutation-invariant aggregation stage (see Lemma 1 and Corollaries 3 and 4) which immediate means that DSPNs can represent any graphic matroid rank function. The expressivity improvement of DSPNs over DSFs means that it is now an open theoretical problem if DSPNs can represent all polymatroid functions or not. Considering the narrow range of submodular functions that DSFs cannot represent, we believe that it is likely that DSPNs indeed can represent any polymatroid function, but we do not have a proof of this at the present time.

We have also, in the present paper, introduced a new learning paradigm (the peripteral loss) for learning DSPNs that does not suffer from the issues that the max-margin learning approach suffers from. In essence, our learning is a form of distillation from an expensive (possibly unoptimizable) target oracle down to a parametric and computationally efficient student (a DSPN). That is, with a peripteral loss, the graded comparison signal $\Delta$ can be any positive/negative value, which could come from a difference in submodular valuations, or could come from a non-submodular function, or could come from a human evaluator (we explain this in detail in Section 1). I.e., the learnt submodular function can, via minimizing the peripteral loss, scale well for all values of any target function (e.g., low valued sets should be given a low value by the learnt function, and high valued set should be given a high value by the learnt function, in order for the peripteral loss to be small). In the present study, we choose $\Delta$ as the difference between expensive-to-optimize facility location function evaluations since it is a widely and successfully used submodular function but that suffers from computational scalability issues when used for optimization (e.g., submodular max) over large datasets.

Another critical novel component of our paper over [23, 13] is that, to the best of our knowledge, our peripteral loss approach is the first to establish a connection between: (I) contrastive learning approaches in representation learning (see Section A.1); (II) pairwise comparison scaling methods widely used in RLHF (and LLM learning, as discussed in the paper, lines 55–81); and (III) submodular learning. We establish connections between these three approaches while at the same time extending regular pairwise comparison approaches to the graded pairwise comparison approach (GPC), which is further novelty. While we do not study contrastive learning or RLHF in our paper, we believe that a broad impact of our work is that our peripteral loss for GPC will be useful for both higher-order contrastive learning methods and for RLHF, but we leave that to future studies as we focus on the submodular learning problem in the present paper (and, as we discuss in our introduction, we believe there is too little attention in the community paid to practical strategies for identifying a good submodular functions for a problem).

Lastly, the experimental results in [23] consider only very small-scale (and entirely offline) summarization experiments (summarizing 100 images down to 10) or synthetic data experiments. By contrast, we consider experiments on a much larger scale (130K images) and apply it towards both offline and online summarization problems in the context of choosing a good set of training data subset points that have not been previously explored — in other words, we instantiate the resulting DSPNs on entirely held-out data, so not only do DSPNs generalize to different sets and set sizes of a given ground set $V$ but DSPNs generalize to entirely different ground sets $V'$ (meaning $V' \cap V = \emptyset$).

So in sum, the present paper offers enormous novelty and difference over the two previous DSF papers [23, 13].

## D  Weighted Matroid Rank Aggregation for DSFs

Vanilla DSFs use modular functions (sum operator) for permutation invariant aggregation. However, as we show in appendix B and definition 6, given a matroid $\mathcal{M} = (V, \mathcal{I})$ a modular function is indeed just a special case of a weighted matroid rank function. Moreover, another common permutation invariant aggregation technique - max-pooling is yet another special case of the weighted matroid rank function. Therefore, with the same notations as in appendix C we define a DSF via a weighted

matroid rank as follows:

$$f_{\mathcal{M}}(A) = W^{(L)} \cdot \Phi_{U^{(L-1)}} \left( W^{(L-1)} \cdot \Phi_{U^{(L-2)}} \left( \ldots W^{(1)} \cdot \Phi_{U^{(0)}} \left( \vec{\max}_{I \in \mathcal{I}} m_{U^{(0)}}(A \cap I) \right) \ldots \right) \right)$$
(17)

where define $\vec{\max}$ as applying maximum operator to every index as follows:

$$\vec{\max}_{I \in \mathcal{I}} m_{U^{(0)}}(A \cap I) = \left( \max_{I \in \mathcal{I}} m_u(A \cap I) \mid u \in U^{(0)} \right)^T$$
(18)

In theorem 13 we show that $f_{\mathcal{M}}$ is a permutation invariant submodular set function. To show that $f_{\mathcal{M}}$ is submodular we first state some properties that will be useful. We re-visit the definitions from [13] for *superdifferential* and *antitone superdifferential* of a concave function which we will use to provide sufficient conditions that will help to show $f$ is submodular.

**Definition 9** (*Superdifferential*). *Let $\phi : \mathbb{R}^d \to \mathbb{R}$ be a concave function; a superdifferential of a concave function is a set defined as follows*

$$\partial \phi(x) = \{ s \in \mathbb{R}^n : f(y) - f(x) \leq \langle s, y - x \rangle, \forall y \in \mathbb{R}^n \}$$

**Definition 10** (*Antitone Superdifferential*). *A concave function $\phi : \mathbb{R}^d \to \mathbb{R}$ is said to have antitone superdifferential if for every $x \leq y$ (where ordering is defined for each dimension) we have a superdifferential $h_x \geq h_y$ for all $h_x \in \partial \psi(x)$ and for all $h_y \in \partial \psi(y)$.*

Using the definitions above, we have the following -

**Lemma 11.** *Given for every layer $\ell$ in a DSF the list of monotone non-decreasing normalized concave functions $\Phi_{U^\ell} = (\phi_u \mid u \in U^\ell)$, $W$ be matrix of non-negative mixture weights. Define $\psi : \mathbb{R}^k \to \mathbb{R}$ as follows*

$$\psi(x) \triangleq W^{(L)} \cdot \Phi_{U^{(L-1)}} \left( W^{(L-1)} \cdot \Phi_{U^{(L-2)}} \left( \ldots W^{(1)} \cdot \Phi_{U^{(0)}} \left( \vec{x} \right) \ldots \right) \right)$$
(19)

*Then $\psi$ is concave and has antitone superdifferential.*

*Proof.* The proof can be done following the similar steps as in Lemma 5.13, Corollary 5.13.1 and Lemma 5.14 from [13]. □

We now re-state Theorem 5.12 in [13] as the follows.

**Lemma 12.** *Let $\psi : \mathbb{R}^k \to \mathbb{R}$ be a monotone non-decreasing concave function and let $g(\vec{A}) \triangleq (g_1(A), g_2(A) \ldots g_k(A))$ be a vector of monotone non-decreasing and normalized submodular (polymatroid) functions. Then if $\psi$ has an antitone super differential, then the set function $f : 2^V \to \mathbb{R}$ defined as $f(A) \triangleq \psi(\vec{g}(A))$ is submodular for every $A \subseteq V$.*

*Proof.* The proof is the same as for Theorem 5.12 in [13]. □

Using the theorem above, we have the following corollary that shows the submodularity of the Deep Submodular Function defined using the weighted matroid rank ($f_{\mathcal{M}}(A)$) as the aggregation function.

**Theorem 13** (Submodularity of $f_{\mathcal{M}}$). *Given a matroid $\mathcal{M} = (V, \mathcal{I})$, $g(\vec{A}) \triangleq \left( \max_{I \in \mathcal{I}} m_u(A \cap I) \mid u \in U^{(0)} \right)^T$ and $\psi$ as in lemma 11, $f_{\mathcal{M}}(A) = \psi(\vec{g}(A))$ is submodular $\forall A \subseteq V$. Moreover, $f_{\mathcal{M}}(A)$ is permutation invariant on the elements of set $A$.*

*Proof.* This is an application of lemma 11 into lemma 12 using the knowledge that weighed matroid rank are polymatroid functions. Permutation invariance of $f_{\mathcal{M}}$ it can be inferred from the permutation invariance of weighted matroid rank function from lemma 1 □

This leads us to the fact that any DSPN, assuming that $w = (w_{\mathfrak{p}}, w_{\mathfrak{r}})$ are valid (specifically, that $w_{\mathfrak{r}} \geq 0$) is submodular.

**Theorem 2** (A DSPN is monotone non-decreasing submodular). *Any DSPN $f_w(A) = \psi_{w_{\mathfrak{r}}}(h_{w_{\mathfrak{p}}}(A))$ so defined is guaranteed to be monotone non-decreasing submodular for all valid values of $w$.*

*Proof.* The output of each pillar is a $d$-dimensional non-negative vector. The aggregation via any weighted matroid rank of the pillar outputs for any set $A \subseteq V$ is also a $d$-dimensional non-negative vector and moreover, each element of this resulting aggregated vector is itself a polymatroid function thanks to the fact that a weighted matroid rank function is a polymatroid function. When we compose a vector of polymatroid functions with a DSF, this preserves polymatroidality by Theorem 5.12 in [13], and thus submodularity. The theorem is hence proven. □

**Corollary 3** (Submodular Preservation of Weighted Matroid Rank Aggregators)**.** *Any weighted matroid rank function for matroid $\mathcal{M} = (V, \mathcal{I})$ with any non-negative vector $m \in \mathbb{R}_+^{|V|}$, when used as an aggregator in a DSPN, will be submodularity preserving.*

*Proof.* This follows immediately from Theorem 2. □

# E    Further discussion on (E,M) pair sampling and dataset construction strategies

Training a DSPN via the peripteral loss requires transferring information from an expensive oracle target function to a learnt DSPN. As mentioned in §section 4, this requires a dataset $\mathcal{D} = \{(E_i, M_i)\}_{i=1}^N$ of pairs of sets to be contrasted with each other where each $E_i, M_i \subseteq V$. The target function in our work is an expensive facility location function, but in general it can be any oracle to provide information in a GPC context to a learner. This means that $\Delta(E|M) = f_t(E) - f_t(M)$. As submodular functions are set functions that tend to produce higher score for a diverse set and a low score for a homogeneous set, the oracle evaluation will be positive if $E$ is considered more diverse than $M$ and will be negative if $M$ is considered more diverse than $E$. For this reason, even if we design sampling strategies that attempt to produce pairs $E, M$ where $E$ is more diverse than $M$, the peripteral loss Eqn. (23) allows for mistakes to be made in the selection of these pairs since the sign of the oracle GPC can switch within the peripheral loss. Hence, our approach is robust to $E, M$ pair label noise in this sense. This also means that our $E, M$ sampling strategy can be done in an automated manner.

Moreover, a set $E$ might be seen as diverse in some cases and homogeneous in other cases. For instance, a set $E$ containing elements from two distinct classes is heterogeneous compared to a set $M$ containing all elements from one class. However, the same set $E$ is homogeneous when compared against a set $E'$ containing all distinct classes, so in this latter case the $(E, M)$ pair would be $(E', E)$. The oracle query offers a GPC of the pair $(E, M)$ providing a score $Score(A, B) \in \mathbb{R}$ depending on which of $E$ and $M$ is more diverse. The pairwise comparison aspect of GPC is one of its powers since we are not asking for absolute scores from the oracle but for comparisons between pairs. This is one of the benefits of pairwise comparison models when querying humans as well (as discussed in the field of psychometrics [74] and this benefit extends to the GPC case as well [61].

For learning a DSPN, an oracle evaluation will require evaluations with an expensive FL function, but in general the oracle need not be submodular at all, and can even be a non-submodular arbitrary dispersion function. Furthermore, as mentioned above, our framework only requires query access to the oracle, we do not need to optimize over it. When the target is non-submodular and we are learning a DSPN, our approach can be seen as finding a projection of a non-submodular function onto the space of DSPNs. A compelling example of this is in the context of dataset pruning, where a target function could be a mapping from pairs of subsets of training data to the difference in validation accuracies that a deep model achieves after being trained on each of the subsets and evaluated on a validation set (we do not test this case in this paper however, but plan to do so in future work). An alternate target function could capture a human's assessment of diversity of a summary, incorporating human feedback into the process of learning a submodular function. Crucially, in both of these examples, the target functions are not directly optimized but are only queried. Our framework may allow us to distill them onto easy-to-optimize functions such as DSPNs that can still exhibit essential and relevant properties of the target.

In general, how the $(E, M)$ pair of subsets are chosen in the learning procedure is absolutely critical, and this is why we have introduced a suite of novel automatic sampling strategies. One of the strategies involves passive sampling of sets via random subsetting and clustering. The other strategy is a new "submodular feedback" method that can be seen as combinatorial active learning since it optimizes over the DSPN as it is being learnt to produce the $(E, M)$ sets, something greatly facilitates learning.

# F Analysis and Further Description of Peripteral Loss

In addition to the motivation for our loss function provided in section 3 here we provide a discussion on its gradient and curvature and how the hyperparameters govern the curvature of the provided loss. First, we re-state the *mother* loss, plot its value for different values of margin, and then follow by equation for its gradient and double-derivative. In the below $\sigma(x)$ refers to sigmoid function, that is, $\sigma(x) = \frac{1}{1+e^{-x}}$ and $\text{sgn}(x)$ refers to the sign of $x$.

The mother loss has the form

$$\mathcal{L}_\Delta(\delta) = |\Delta| \log(1 + \exp(1 - \frac{\delta}{\Delta})), \tag{20}$$

and the corresponding gradient becomes

$$\frac{\partial \mathcal{L}_\Delta(\delta)}{\partial \delta} = -\text{sgn}(\Delta)\sigma\left(1 - \frac{\delta}{\Delta}\right). \tag{21}$$

We first plot the loss and the gradients in Fig. 7. From the plots in Fig. 7 and gradient in Fig. 21, it is clear that if $\Delta$ is (very) positive, then $\delta$ must also be (very) positive to produce a small loss. If $\Delta$ is (very) negative then so also must $\delta$ be to produce a small loss. The absolute value of $\Delta$ determines how non-smooth loss is so we include an outer pre-multiplication by $|\Delta|$ ensures gradient of loss, as a function of $\delta$, asymptotes to negative one (or one when $\Delta$ is negative) in the penalty region, analogous to hinge loss, and thus produces numerically stable gradients.

Having established the "mother" loss $\mathcal{L}_\Delta(\delta)$ we can see that $\Delta$ controls its curvature (the smaller the $|\Delta|$ the higher the curvature). Still, we augment the mother loss with additional hyperparameters in way that produces smoother and more stable and convergent optimization. Firstly, rather than a requiring a margin of one, we introduce a hyperparameter $\tau$ to express this margin whose default value is $\tau = 1$. Second, since the units of the oracle target might be different than the units of the learner, we introduce a unit adjustment parameter $\kappa$ that scales $\Delta$. We also introduce an anti-smoothness parameter $\beta > 0$ (smaller $\beta$ means smoother loss) to further optionally smooth the transition between the linearly increasing loss region and the zero-loss region of $\delta$. That is, since in general the margin could be different from sample to sample, the $\beta$ hyperparameter provides global control over the curvature of the loss in a way that produces smoother, more stable, and convergent optimization. Lastly, if $\Delta$ gets small, to avoid finite-precision numerical problems, we add a small signed value $\epsilon \, \text{sgn}(\Delta)$ for $\varepsilon \geq 0$ which slightly pushes the denominator away from zero depending on the sign of $\Delta$. This yields $\mathcal{L}_{\Delta;\beta,\tau,\kappa,\varepsilon}(\delta)$ which is defined as:

$$\mathcal{L}_{\Delta;\beta,\tau,\kappa,\varepsilon}(\delta) \triangleq \frac{|\kappa\Delta + \varepsilon \, \text{sgn}\,\Delta|}{\beta} \log(1 + \exp(\beta(\tau - \frac{\delta}{\kappa\Delta + \varepsilon \, \text{sgn}\,\Delta}))) \tag{22}$$

There is one last issue with the above we must address, namely when $\Delta$ becomes extremely small or zero. This is perfectly reasonable since it expresses the oracle's indifference to $E$ vs. $M$. We address this with a $\tanh$-based "gating" function that detects when $\Delta$ is very small and gates towards a secondary loss that penalizes the absolute value of $\delta$. The final peripteral loss $\mathcal{L}_{\Delta;\alpha,\beta,\tau,\kappa,\varepsilon}(\delta)$ thus has the form:

$$\mathcal{L}_{\Delta;\alpha,\beta,\tau,\kappa,\varepsilon}(\delta) \triangleq \tanh\left(\frac{\alpha}{|\kappa\Delta|}\right)|\delta| + \left(1 - \tanh\left(\frac{\alpha}{|\kappa\Delta|}\right)\right)\mathcal{L}_{\Delta;\beta,\tau,\kappa,\varepsilon}(\delta) \tag{23}$$

The rate of the gating is determined by $\alpha \geq 0$, and since $\frac{\alpha}{|\kappa\Delta|} \geq 0$, the gate switches to the absolute value $|\delta|$ whenever $\Delta$ gets very small. Thus if the oracle target is indifferent to $E$ vs. $M$ this encourages the learner also to be indifferent. Despite these hyperparameters, the loss gradient is still independent of the margin and $\beta$ making it useful for fine-tuning a learning rate

Considering only following term (which does not have the gating term),

$$L_{\Delta;\beta,\tau,\kappa,\varepsilon}(\delta) = \frac{|\kappa\Delta + \varepsilon \, \text{sgn}\,\Delta|}{\beta} \log(1 + \exp(\beta(\tau - \frac{\delta}{\kappa\Delta + \varepsilon \, \text{sgn}\,\Delta}))) \tag{24}$$

the corresponding gradient will be:

$$\frac{\partial L_{\Delta;\beta,\tau,\kappa,\varepsilon}(\delta)}{\partial \delta} = -\text{sgn}(\Delta)\sigma\left(\beta\left(\tau - \frac{\delta}{\kappa\Delta + \varepsilon \, \text{sgn}\,\Delta}\right)\right). \tag{25}$$

We plot the gradients in appendix F for two offsets, namely $\tau = 1$ and $\tau = 5$. From these plots, it is clear that $\beta$ provides control over the smoothness of $L_{\Delta;\beta,\tau,\kappa,\varepsilon}(\delta)$ where the smaller the $\beta$ the smaller the curvature (without impacting the gradient magnitudes). The importance of low curvature of the loss objective has been studied from the optimization stability [30] as well as from the lens of robustness in deep learning [95, 27]. Similarly, we can also see that $\tau$ merely provides an offset in the gradient, which, however, can nevertheless be important for the gradient magnitude while we are optimizing.

Bringing back in the gating term (again, to account for corner cases where $\Delta$ is very small), we have

$$L_{\Delta;\alpha,\beta,\tau,\kappa,\varepsilon}(\delta) = \tanh\left(\frac{\alpha}{|\kappa\Delta|}\right))|\delta| + (1 - \tanh\left(\frac{\alpha}{|\kappa\Delta|}\right))\mathcal{L}_{\Delta;\beta,\tau,\kappa,\varepsilon}(\delta), \tag{26}$$

and the corresponding gradient will be:

$$\frac{\partial L_{\Delta;\alpha,\beta,\tau,\kappa,\varepsilon}(\delta)}{\partial \delta} = \tanh\left(\frac{\alpha}{|\kappa\Delta|}\right)) \operatorname{sgn}(\delta) + (1 - \tanh\left(\frac{\alpha}{|\kappa\Delta|}\right))\frac{\partial L_{\Delta;\beta,\tau,\kappa,\varepsilon}(\delta)}{\partial \delta}. \tag{27}$$

Note that for $\delta = 0$ the function is indeed non-differentiable, however, we can have any subgradient between $[-1, 1]$. Similar to PyTorch [78] we choose to have the subgradient to be 0, as $|\delta|$ is also 0 at that point. In practice, we do not encounter any instabilities due to this, and gate value throughout our experiments remains very small.

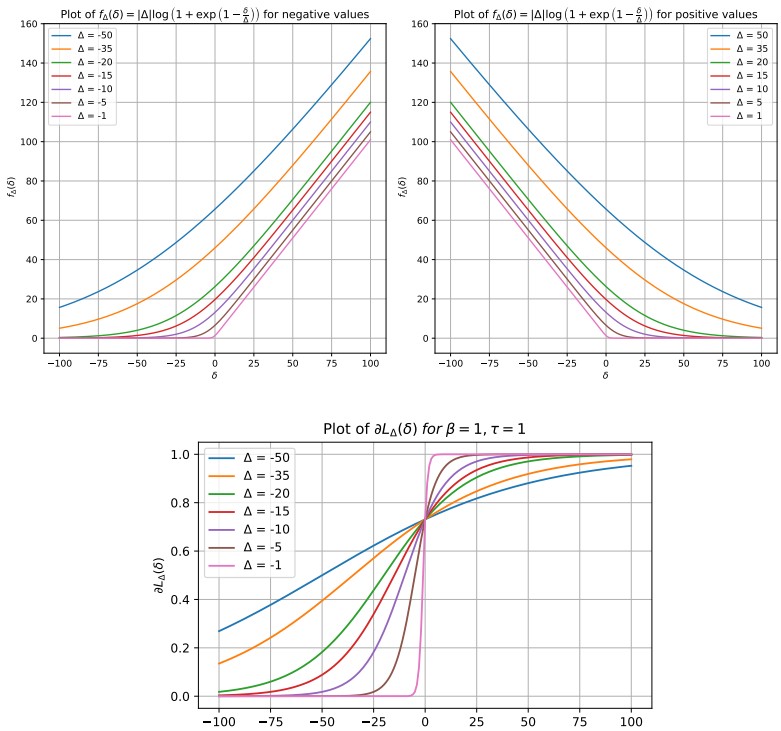

Figure 7: Peripteral loss of $\delta$ for different values and sign of margin $\Delta$ and the corresponding gradient.

### F.1  Final Extended Loss

Building and extending Section 3.2 using the peripteral loss along with its hyperparameters, and with all of the individual loss components described from Section 3.1, the final total loss $\mathcal{L}_{\text{tot}}(f_w; \Delta(E|M), \delta_{f_w}(E|M))$ for an $E, M$ pair becomes:

$$\mathcal{L}_{\text{tot}}(f_w; \Delta(E|M), \delta_{f_w}(E|M)) = \mathcal{L}_{\Delta(E|M);\alpha,\beta,\tau,\kappa,\varepsilon}(\delta_{f_w}(E|M)) + \mathcal{L}_{\text{aug}}(f_w; E \cup M, E' \cup M')$$
$$+ \mathcal{L}_{\text{redn}}(f_w; E, E') + \mathcal{L}_{\text{redn}}(f_w; M, M') \tag{28}$$

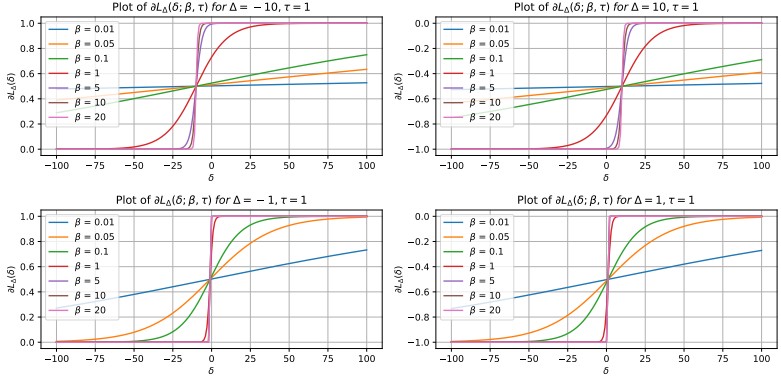

Figure 8: $\tau = 1$.

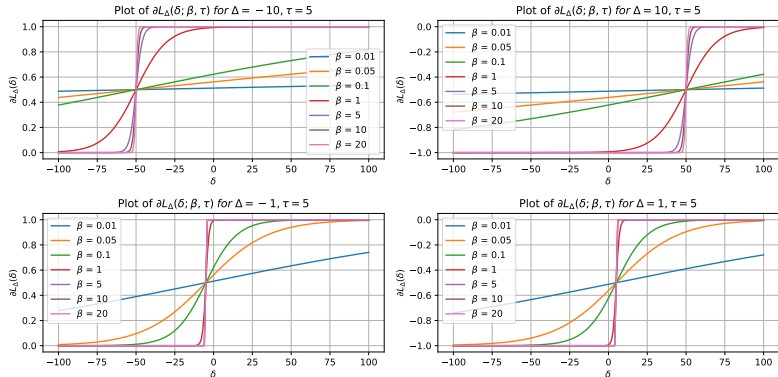

Figure 9: $\tau = 5$

With a dataset $\mathcal{D} = \{(E_i, M_i)\}_{i=1}^N$ of pairs, the total risk is:

$$\hat{\mathcal{R}}(w; \mathcal{D}) = \frac{1}{N} \sum_{i \in [N]} \mathcal{L}_{\text{tot}}(f_w; \Delta(E_i|M_i), \delta_{f_w}(E_i|M_i)). \tag{29}$$

# G Peripteral Networks and Training Flow

Figure 1 in the main paper depicts the DSPN architecture. In fact, we see two copies of a DSPN presented in Figure 1, the left one instantiated for the $E$ set and the right one instantiated for the $M$ set. Each DSPN consists of a set of pillars which transform a number of objects (e.g., images corresponding to a set) into a non-negative embedding space. The reason for the space to be non-negative is to ensure that the subsequent transformations can preserve submodularity, in the same way that a concave composed with a modular function requires the modular function to be non-negative to preserve submodularity. The figure above shows that object space has dimension nine (the number of red nodes on the bottom of each pillar) and embedding space having dimension seven (the seven orange nodes at the top of each pillar). This DSPN has five pillars which means that it is appropriate for an $E$ or $M$ set of size five. The outputs of the pillars are aggregated using a submodular-preserving permutation invariant aggregator (such as a weighted matroid rank function) but in the experiments in this paper, we use a simple uniform matroid which corresponds to a summation aggregator. Note that each scalar output of each pillar is aggregated separately and after aggregation we get a resulting seven-dimensional vector which is input to a deep submodular function (DSF). We consider the aggregation step and the final DSF to be the "roof model.", highlighted with the blue triangle above. The word **peripteral** is an adjective (applicable typically to a building) which means having a single row of pillars on all sides in the style of the temples of ancient Greece. This is reminiscent of the above DSPN structures which is the reason for this name (see figure10).

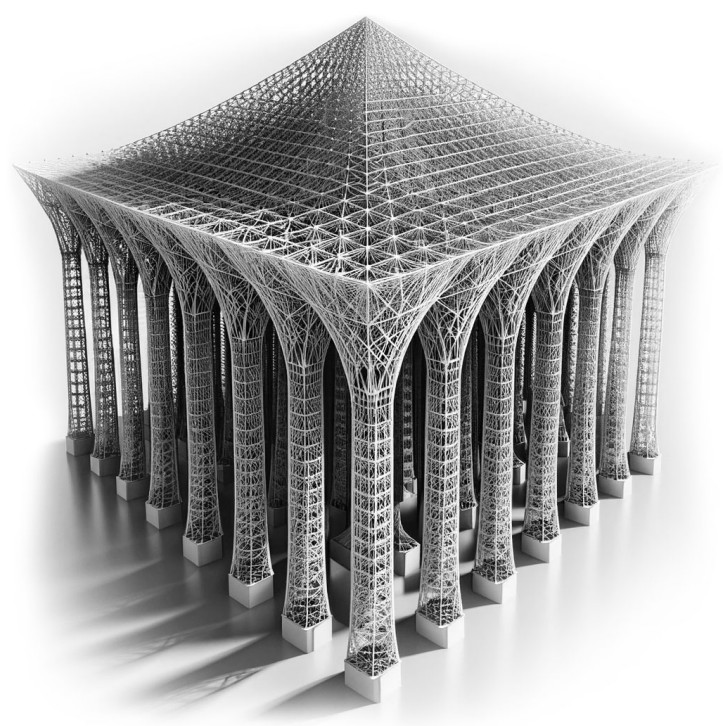

Figure 10: A "peripteral neural temple" showing a Greek/Roman style peripteral structure with many pillars rendered as a form of neural temple, where each of the pillars, the submodular preserving and permutation-invariant aggregation, and the roof are all seen as having an underlying neural network graphical structure. This figure was generated using GPT-4 after an enormous amount of trial and error including many seed images, and then it was further edited in Adobe Photoshop.

To train a DSPN, we need two sets, an $E$ (hEterogeneous) set and an $M$ (hoMogeneous) set. Each of the $E$ and $M$ sets of the $(E, M)$ pair is presented to a DSPN instantiation being learnt and which is governed, overall by the parameters $w$. Note that $w$ consists both of the parameters for the DSPN pillars as well as the parameters for the DSPN roof. The parameters for all instances of the pillar are shared over all pillars, similar to how Siamese networks share (or tied) parameters between the two transformations. Here, however the number of pillars in the DSPN grows or shrinks depending on the sizes $|E|$ and $|M|$ of the sets $E$ and $M$ respectively, but regardless of the size of $|E|$ or $|M|$ the number of the parameters of the pillars stays fixed. These sets are acquired in a variety of different ways, and this includes heuristics on the data (e.g., clusterings, random sets) as well as submodular feedback strategy which optimizes the DSPN as it is being learnt to ensure that what the DSPN thinks has high or low value agrees with the target oracle.

## H   Additional Experimental Details

**Baseline Loss Functions**   To assess the efficacy of our peripteral loss, we consider two popular baselines to learn from pairwise comparisons. Note that we only replace our peripteral loss function with the baseline and we do not remove any necessary regularizers such as $\mathcal{L}_{redn}$ and $\mathcal{L}_{aug}$. Using the same notations as the main paper, we define the regression loss $\mathcal{L}_{\text{regr}}$ and $\mathcal{L}_{\text{marg}}$ in equations 30 and 31.

$$\mathcal{L}_{\text{regr}}(f_w; \Delta(E|M), \delta_{f_w}(E|M)) = (\delta_{f_w}(E|M) - \Delta(E|M))^2 \tag{30}$$

$$\mathcal{L}_{\text{marg}}(f_w; \Delta(E|M), \delta_{f_w}(E|M)) = \max(\Delta(E|M) - \delta_{f_w}(E|M)), 0) \tag{31}$$

Given the dataset $\mathcal{D} = \{(E_i, M_i)\}_{i=1}^N$ of pairs, we have the following empirical risks for regression (Eq. 32) and margin (Eq. 33).

$$\hat{\mathcal{R}}_{\text{regr}}(w; \mathcal{D}) = \frac{1}{N} \sum_{i \in [N]} (\mathcal{L}_{\text{regr}}(f_w; \Delta(E_i|M_i), \delta_{f_w}(E_i|M_i)) + \mathcal{L}_{\text{aug}}(f_w; E \cup M, E' \cup M') \tag{32}$$
$$+ \mathcal{L}_{\text{redn}}(f_w; E, E') + \mathcal{L}_{\text{redn}}(f_w; M, M')$$

$$\hat{\mathcal{R}}_{\text{marg}}(w; \mathcal{D}) = \frac{1}{N} \sum_{i \in [N]} \mathcal{L}_{\text{marg}}(f_w; \Delta(E|M), \delta_{f_w}(E|M)) + \mathcal{L}_{\text{aug}}(f_w; E \cup M, E' \cup M') \tag{33}$$
$$+ \mathcal{L}_{\text{redn}}(f_w; E, E') + \mathcal{L}_{\text{redn}}(f_w; M, M')$$

**Hyperparameters and Sensitivity Analysis**  We train every DSPN on 2 NVIDIA-A100 GPUs with a cyclic learning rate scheduler [94], using Adam optimizer [47]. To ensure non-negative parameters in DSF, we perform projected gradient descent, where we apply projection on $\psi_{w_{\text{r}}}$ to the non-negative orthant after every gradient descent step.

We list the hyperparameters we use for the peripteral loss in table 2 and table 3. We observed that $\beta$ and $\tau$ are the most salient hyperparameters since they control the curvature and hinge of the loss respectively. The benefits of smaller loss curvature have been widely studied from optimization stability in deep learning [30, 95, 27] and our low value of $\beta$ supports previous work. $\alpha$ parameter controls the tanh gate, the smaller the value of alpha the more weight is given to the main part of our objective function. Since we did not encounter corner cases where $\Delta$ is very small, neither $\epsilon$ nor $\alpha$ were found to be particularly influential.

| | $\alpha$ | $\beta$ | $\kappa$ | $\tau$ | $\epsilon$ |
|---|---|---|---|---|---|
| Imagenette | $1e-5$ | $0.5$ | $1$ | $10$ | $1e-15$ |
| Imagewoof | $1e-5$ | $0.5$ | $1$ | $10$ | $1e-15$ |
| CIFAR100 | $1e-5$ | $0.01$ | $1$ | $10$ | $1e-15$ |
| Imagenet100 | $1e-5$ | $0.01$ | $1$ | $10$ | $1e-15$ |

Table 2: Peripteral loss hyperparameters used for each dataset. $\lambda_1$ and $\lambda_2$ correspond to the coefficients used in the augmentation regularizer (Eq. 2, while $\lambda_3$ and $\lambda_4$ are the coefficents in (Eq. 3)

| | $\lambda_1$ | $\lambda_2$ | $\lambda_3$ | $\lambda_4$ |
|---|---|---|---|---|
| Imagenette | $0.25$ | $0.01$ | $0$ | $0$ |
| Imagewoof | $0.25$ | $0.01$ | $0$ | $0$ |
| CIFAR100 | $0.25$ | $0.01$ | $1e-5$ | $5e-4$ |
| Imagenet100 | $0.25$ | $0.01$ | $0$ | $0$ |

Table 3: Loss coefficients used for each dataset.

We search over $\beta \in \{0.01, 0.1, 0.5\}$ and $\tau \in \{1, 5, 10\}$ and report the normalized target evaluation for Imagenet100 results in Table 4 and Table 5. We select which hyperparameter configuration to use based on normalized FL evaluation which is described in Figure 4. We find that DSPN can work reasonably well even without tuning $\beta$ and $\tau$, but a hyperparameter search should be done for peak performance. Regarding the loss coefficients, we fix the $\lambda_1 = 0.25$ and $\lambda_2 = .01$ respectively across all datasets, where the range is chosen such that each loss value is commensurate with the other. This way, none of the loss/regularization terms solely drive the optimization. For $\lambda_3$ and $\lambda_4$ we decided the range to be $0, 1e-5, 1e-3$ and $0, 5e-4, 5e-2$. For Imagnette/Imagewoof/Imagenet-100 we did not see significant changes in performance by including non-zero $\lambda_3$ and $\lambda_4$. In fact, even for CIFAR-100 not having these does not imply DSPN will fail to produce good summaries, it only results in a marginal drop.

**Dataset Construction**  We compute the passive $(E, M)$ pairs offline to improve computational efficiency. For the actively sampled pairs, we generate $(E, M)$ pairs every $K$ epochs where an epoch is defined as a full pass over all of the offline pairs. For all the datasets we consider, $K = 15$.

|               | k=10 | k=20 | k=30 | k=40 | k=50 | k=60 | k=70 | k=80 | k=90 | k=100 |
|---------------|------|------|------|------|------|------|------|------|------|-------|
| $\beta = 0.01$ | 0.88 | 0.89 | 0.92 | 0.91 | 0.91 | 0.91 | 0.92 | 0.92 | 0.92 | 0.92 |
| $\beta = 0.1$  | 0.88 | 0.88 | 0.90 | 0.90 | 0.90 | 0.92 | 0.91 | 0.92 | 0.92 | 0.92 |
| $\beta = 0.5$  | 0.90 | 0.89 | 0.90 | 0.92 | 0.92 | 0.92 | 0.92 | 0.93 | 0.92 | 0.92 |

Table 4: Sensitivity of $\beta$ based on normalized FL evaluation.

|              | k=10 | k=20 | k=30 | k=40 | k=50 | k=60 | k=70 | k=80 | k=90 | k=100 |
|--------------|------|------|------|------|------|------|------|------|------|-------|
| $\tau = 1$   | 0.84 | 0.84 | 0.85 | 0.87 | 0.88 | 0.88 | 0.88 | 0.89 | 0.90 | 0.90 |
| $\tau = 5$   | 0.90 | 0.88 | 0.88 | 0.89 | 0.90 | 0.90 | 0.90 | 0.91 | 0.91 | 0.92 |
| $\tau = 10$  | 0.90 | 0.89 | 0.90 | 0.92 | 0.92 | 0.92 | 0.92 | 0.93 | 0.92 | 0.92 |

Table 5: Sensitivity of $\tau$ based on normalized FL evaluation.

**Long-tailed Dataset Construction**    For the results presented in Sections 5.2, we create pathologically imbalanced datasets from sets of images that the DSPN never encountered during training. Specifically, we add duplicates for each image in a class; the number of duplicates varies per class and is drawn from a Zipf distribution [116]. We show the final distributions of the imbalanced dataset in Figure 11.

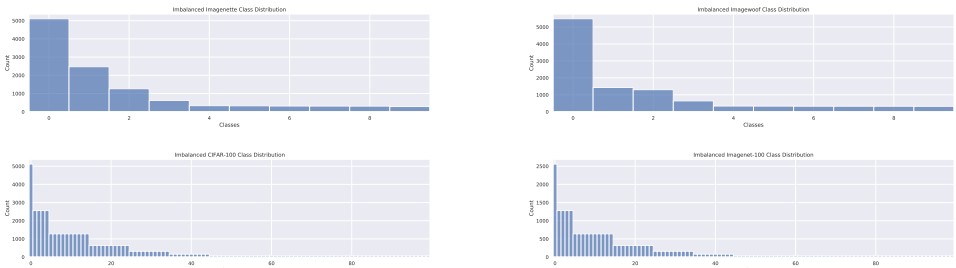

Figure 11: Imbalanced dataset distributions

# I   Qualitative Analysis of Learnt Features

In Table 1, we show that imparting DSPNs the ability to learn features is crucial for successful training. Feature-based functions and DSFs work best when features correspond to some bag-of-words or count like attribute. For example, [60, 62] both use TF-IDF based feature extractors to instantiate feature-based functions. More recently, [51] learns "additively contributive" (AC) features with an autoencoder by restricting all weights in the initial layer of the decoder to the positive orthant. Unfortunately, universal features for images procured from modern pretrained models such as CLIP do not necessarily have this property.

We show that DSPNs learn bag-of-words features in Figure 12, by exploring the preferences of individual DSPN/CLIP features. For visualization purposes, we randomly select 10 features from the model in consideration and use each feature to rank the images from a completely held out set. The top 10 images selected by any of the DSPN features appear very homogeneous and are strongly correlated to class, while images with the lowest feature values (typically with feature value 0) are practically indistinguishable from a random subset of images. In other words, a high value indicates the *presence* of some attribute while a low value indicates its *absence*. When we perform a similar qualitative analysis upon CLIP features, we find that both the high valued sets and low valued sets have some weak correlation to class. CLIP features cannot be interpreted in the same way as DSPN because each feature corresponds to a coordinate in some higher dimensional space. Our results show that features that encapsulate count-like attributes are essential for learning a good DSF.

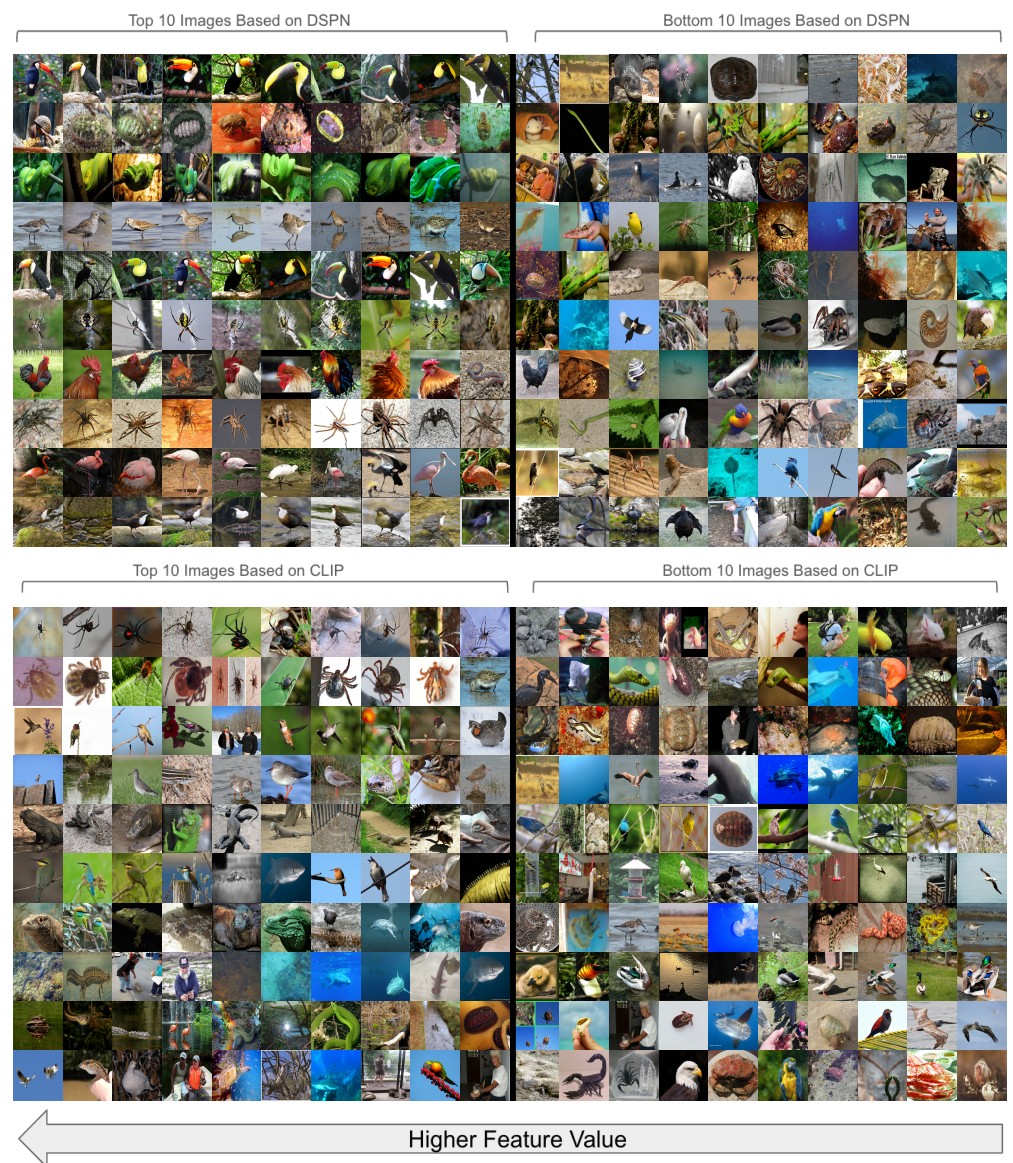

Figure 12: **Qualitative Assessment of Learnt Features on Imagenet100.** For each row, we use a randomly selected feature of either DSPN or CLIP to rank all images in the held out set and visualize the highest and lowest ranked samples. Each DSPN feature corresponds to some count-like property that is correlated to class, evident from the fact that very few distinct classes are present in the highly ranked images.

| k | DSPN | DSPN (No Redun) | Random |
|---|---|---|---|
| 10 | 0.91 | 0.88 | $0.65 \pm 0.04$ |
| 20 | 0.93 | 0.89 | $0.67 \pm 0.03$ |
| 30 | 0.91 | 0.89 | $0.67 \pm 0.02$ |
| 40 | 0.90 | 0.88 | $0.68 \pm 0.02$ |
| 50 | 0.90 | 0.88 | $0.68 \pm 0.02$ |
| 60 | 0.89 | 0.88 | $0.68 \pm 0.02$ |
| 70 | 0.88 | 0.88 | $0.68 \pm 0.02$ |
| 80 | 0.88 | 0.87 | $0.69 \pm 0.02$ |
| 90 | 0.88 | 0.87 | $0.69 \pm 0.02$ |
| 100 | 0.88 | 0.87 | $0.70 \pm 0.02$ |

Table 6: Effect of setting $\lambda_3 = \lambda_4 = 0$ on normalized FL evaluations on CIFAR100

## J   Limitations

A limitation of DSPNs is that they may not be able to produce effective summaries that are significantly larger than the largest sets that they were trained on. In our experiments, we do observe that the learnt DSPN produces high quality summaries of size 500 though they were not trained on sets larger than 100. However, further investigation is needed to determine if DSPNs can continue to produce high quality summaries with significantly larger cardinalities.

