# OpenReview forum: "Deep Submodular Peripteral Networks"
_NeurIPS.cc/2024/Conference — NeurIPS 2024 spotlight_

### Official Review · Reviewer_Ltg4 · 2024-07-11

**Soundness:** 3
**Presentation:** 2
**Contribution:** 3
**Rating:** 7
**Confidence:** 4

**Summary:**

The authors identify two open problems in machine learning and provide novel solutions to them. The first problem is that even though the submodular functions show up in numerous applications, learning submodular functions through DNNs remain unpractical. Their proposed solution to this issue is their new architecture called Deep Submodular Peripteral Networks (abbreviated as DSPNs). The other identified problem is the lack of graded pairwise preferences (GPCs) oracles. In its essence, it is the type of oracle where given two sets of elements, a score with a scalar value is returned. The sign of this score denotes which set is preferred over the other set and the absolute value of the score denotes the magnitude of this preference. To reflect the effects of this oracle better, they introduce a new loss function called peripteral loss. They evaluate DSPNs ability to learn a target function (a facility location function in the case of their experiments) against different baseline algorithms. They also demonstrate the effects of using different losses when training DSPNs.

**Strengths:**

Considering the success of supervised learning from comparisons in domains such as healthcare etc., it is no surprise that learning from graded pairwise comparisons is a significant oracle selection. I can see it being very useful in team selection problems where we want to predict the outcome of a competition. The DSPN structure is original. Majority of the ideas conveyed clearly.

**Weaknesses:**

- I have some concerns related to the organization of the paper. I am aware the page limitations can be annoying sometimes. Still, I do think that Figure 13 should be included in the main body of the paper. I think it would help readers understand the paper better and the main body of the paper would be more standalone that way. Maybe just a 2+1+1 or 2+1+2 layer version of Fig. 13 as you describe in your experiments.

- The introduction mentions E, M sets but we don't know what they stand for until later. You may consider mentioning them as heterogeneous and homogeneous sets before your main contributions.

Minor editorial comments:
- Appendix or Figure names are not properly capitalized at some places e.g. Line 810.
- Some figures refer the reader to the appendix without denoting they are in the appendix.
- On line 96-97, the adjectives need commas in between.
- Line 844, multiple "also"s

**Questions:**

- For my understanding, can we interpret DSPNs as enhanced versions of DSFs with permutation invariance?
- Where do weighted matroid rank functions stand in relation to threshold potentials (Stobbe and Krause [95])?
- Isn't Proposition 8 a bit obsolete? Aren't all set functions characterized by their permutation-invariance hence the submodular set functions as well?

**Limitations:**

Yes, the authors discuss the limitations of their work in Appendix J.

---

> ### Author Rebuttal · Authors · 2024-08-07
>
> Firstly, thank you for your review and your comments. We will attempt in the next version of the paper to address all of them. Detailed comments follow.
>
>
> **Organization**
> - We agree that Figure 13 ideally belongs in the main body. If the paper is accepted, for the final camera ready we are allotted one additional page that can fit Figure 13 as well as other motivating and other material.
>
>
> **Descriptions of E,M**
> - Yes, we can describe them as the heterogeneous and homogeneous sets right when the notation is first introduced.
>
> **editorial comments**
> - These typos will all be fixed in the next version. Thank you for pointing them out.
>
> **DSPNs as enhanced versions of DSFs with permutation invariance**
> - DSPNs are actually even more than this since DSPNs correspond to a larger sub-family of submodular functions than DSFs, as we prove in our paper. It may be that DSPNs can represent any submodular function, but we do not yet have a proof for that, but it is currently a conjecture.
>
>
> **DSPNs vs threshold potentials (Stobbe and Krause [95])**
> - This is a good question. In the DSF theory paper (Bilmes & Bai 2017), it was shown that DSFs extend the family of sums of concave composed with modular functions (SCCMs). The threshold potential functions of Stobbe & Krause are instances of SCCMs and hence DSFs correspond to a larger family than threshold potentials. DSPNs, moreover, correspond to a still larger sub-family of submodular functions than SCCMs as mentioned above. Hence, DSPNs are a larger family than those of Stobbe and Krause. Incidentally, Stobbe and Krause refer to SCCMs as "decomposable submodular functions" but we prefer the name "SCCMs" or "feature based functions" in order to avoid the overloading of and potential confusion between that and the notion of "decomposable graphs" in graphical models and chordal graph theory. The reason for this potential confusion is that the graph notion of decomposability can apply to a submodular function as well separately from if the submodular function is representable as an SCCM or not.
>
> **Proposition 8**
> - We consider a main theoretical result of the paper that the weighted matroid rank functions in the middle of the DSPN preserve submodularity and permutation invariance of the DSPN. We include proposition 8 only to be complete in case there is any question, but in the next version of the paper we can clarify this.
>
> Again, thank you very much for your review and your time!

---

> > ### Comment · Reviewer_Ltg4 · 2024-08-08
> >
> > Thank you for addressing my comments! I will keep my score. Good luck!

---

> > > ### Author Response · Authors · 2024-08-12
> > >
> > > We thank the reviewer for their response!

---

### Official Review · Reviewer_W4dW · 2024-07-12

**Soundness:** 3
**Presentation:** 3
**Contribution:** 3
**Rating:** 7
**Confidence:** 2

**Summary:**

This paper proposes a new framework to learn submodular functions.  Specifically, the paper proposes a new parametric family of submodular functions using neural networks. Then, the paper proposes training such networks with a new loss function that is based on graded pairwise comparisons.

**Strengths:**

I think the paper addresses an important problem of learning/distilling submodular functions in a scalable way. The paper also conducts thorough experiments comparing their approach with baselines.

The paper is well-written and easy to follow. The paper does a good job discussing related and prior work.

**Weaknesses:**

This paper is outside my area of expertise, and as such I don't think I can offer constructive feedback.

**Questions:**

None.

**Limitations:**

Yes.

---

> ### Author Rebuttal · Authors · 2024-08-07
>
> We nonetheless wish to thank you for your review and time. We are glad that you found the paper well-written and easy to follow.

---

> > ### Comment · Reviewer_W4dW · 2024-08-09
> > **Response to Rebuttal**
> >
> > Thank you for the response.

---

### Official Review · Reviewer_YWZB · 2024-07-14

**Soundness:** 3
**Presentation:** 2
**Contribution:** 3
**Rating:** 6
**Confidence:** 2

**Summary:**

The paper introduces Deep Submodular Peripteral Networks (DSPNs), a novel parametric family of submodular functions designed to address practical learning methods for submodular functions and graded pairwise comparisons (GPC). To learn DSPNs, the paper also introduces a new GPC-style “peripteral” loss, which leverages numerically graded relationships between pairs of objects, extracting more nuanced information than binary comparisons. Finally, the authors demonstrate DSPNs’ efficacy in learning submodularity from a
costly target submodular function showing superiority both for experimental design and online streaming applications.

**Strengths:**

- The paper introduced Deep Submodular Peripteral Networks (DSPNs), a new parametric family of submodular functions. Since the computational cost for querying a DSPN is independent of dataset size, their approach can learn practical submodular functions scalably.

- The paper also proposed a new GPC-style “peripteral” loss to successfully learn DSPNs. As introduced, this loss has many applications such as learning a reward model in the context of RLHF.

**Weaknesses:**

- It lacks a theoretical guarantee for the proposed peripteral loss. What happens if the target oracle does not exist?

**Questions:**

- Can the authors explain intuitively how the peripteral loss is used to learn a reward model in RLHF?

**Limitations:**

Yes

---

> ### Author Rebuttal · Authors · 2024-08-07
>
> Thank you for your review and time. We address your questions below.
>
>
> **Theoretical guarantee for the proposed peripteral loss**
> - In our submission, we have included theoretical results regarding the guarantee that submodularity is retained by the use of the permutation invariant stage of a DSPN via the use of weighted matroid rank functions. There is still an open theoretical problem if a DSPN can perfectly represent (or arbitrarily closely approximate) *any* submodular function which currently is a conjecture. There is also the following conjecture: If the resulting peripteral loss, once trained, falls below a given threshold, then for each pair E,M with \Delta(E|M) > 0 (respectively \Delta(E|M) < 0), the volume in embedding space of the convex hull of the points in E will be greater (resp. smaller) than the volume in embedding space of the convex hull of the points in M.
>
> **Target oracle does not exist**
> - In such case, if an approximate oracle exists than we should still be able to train. Note that the oracle only needs to be able to be queried, we do not need to optimize over the oracle which means that non-submodular oracles (such as non-submodular functions or humans) can be the oracle. This segways to your third comment.
>
> **Peripteral loss is used to learn a reward model in RLHF**
> - This is briefly outlined on lines 45-59. The key idea is that RLHF could be used as normal via, say, PPO (proximal policy optimization) but the key difference is that the human providing the feedback would need to provide more than just a binary value of if A is better than B or B is better than A, rather a "graded" preference would need to be given by the computer saying how much is A better than B (or B better than A). This feedback is then represented via our $\Delta(E|M)$ score except here rather than scoring the preference for an $E$-set over an $M$ set (which would not be present), it would instead score two LLM-responses $A$ and $B$ relative to each other and the error (which is the peripteral discrepancy between the human's preference score and the model's difference score) would then propagate back to an LLM as in PPO via the peripteral loss. We hope to be able to pursue this research direction in the future.
>
> Again, thanks very much for your review and comments.

---

### Official Review · Reviewer_fthD · 2024-07-16

**Soundness:** 3
**Presentation:** 1
**Contribution:** 4
**Rating:** 7
**Confidence:** 3

**Summary:**

The paper introduces deep submodular peripteral networks (DSPNs) and a graded pairwise preferences (GPC)-style peripteral loss. It shows that DSPNs are effective in learning submodularity from a target function.

**Strengths:**

- The construction of the submodular function, i.e. DSPNs are interesting as they assure the submodularity.
- The loss function’s construction using contrastive sets is also interesting and reasonable.
- The experiments demonstrate that DSPNs combined with the peripteral loss are effective.

**Weaknesses:**

The paper’s organization needs improvement. Many important concepts, such as submodular functions, FC functions, and GPC, lack definitions (even informal) when they first appear. Additionally, there is no motivation or detailed application of submodular functions in machine learning, making it difficult for readers to understand at the beginning.

Minor: Page 5, line 209, “then so also must”.

**Questions:**

1. I understand that due to page limits, a large portion of the content is included in the appendix. However, I suggest providing a short sentence with informal definitions or references to the definition locations for important concepts so that readers can quickly grasp their meanings and follow the text more naturally. For example, on line 61, what are graded pairwise preferences? Additionally, where is the formal definition of a submodular function? What are the definitions of a matroid and a matroid rank function?
2. I recommend adding some motivation and detailed applications about how submodular functions are used in machine learning and why they are important. Explicitly explaining where and how submodular functions can be applied in the introduction would be beneficial, instead of generally stating that they have been used in several areas and studied in several papers.
3. Could the output of the function be a vector of a few dimensions instead of a scalar, where each dimension represents different aspects of comparison, or can be scores judged by different people? Can you provide some insights on whether this scenario is reasonable and if so, whether your framework can be applied to this setting?
4. In Section 5, what is the target FL and how is it labeled/created? I suggest state explicitly on the task (e.g., image summarization) in the main text.

**Limitations:**

The authors have adequately addressed the limitations.

---

> ### Author Rebuttal · Authors · 2024-08-07
>
> We thank the reviewer for their feedback and questions, and are glad that they found our work to be convincing. We address each question in turn.
>
> **More definitions in the main body**
> - Yes, in the next version of the paper we will add more such definitions in the main body. According to the Neurips2024 call for papers, if the paper is accepted, we will be granted an additional content page that we can use for this purpose, including as you suggest GPC, and the basic definitions of and motivations for submodularity and matroids.
>
> **Output a vector rather than a scalar**
> - This is an interesting question that we didn't pursue in the present paper, but there is no reason that our approach would not generalize to vector outputs, assuming an oracle queryable teacher is available. Given the opportunity, we will discuss this possibility in the next version.
>
> **What is target FL?**
> - While the oracle need not be an FL (facility location) function or even a submodular function, in the present work we use an expensive to optimize but easy to query target FL function as the oracle, and we transfer from this target FL to the learnt DSPN model. In our present work, each target FL is created by computing a real-valued feature vector for each sample and then computing a non-negative similarity between each such vector. We used a CLIP ViT encoder to encode input images, and a tuned RBF kernel to construct similarities for our target FL. Note that the target FL construction is rather expensive (quadratic) and also does not generalize to held-out data, but this is a key point. That is, the paper shows how we distill from this expensive target FL to a cheaper and generalizable DSPN.
>
>
> **Main Task**
> - Yes, we will do this in the revised version.
>
> Again, thanks for your review and comments!

---

> > ### Comment · Reviewer_fthD · 2024-08-07
> > **After Rebuttal**
> >
> > Thanks for the reply. The authors have addressed my questions. I will keep my score.

---

> > > ### Author Response · Authors · 2024-08-12
> > >
> > > We thank the reviewer for their response!

---

### Author Rebuttal · Authors · 2024-08-07

We wish to thank all of the reviewers for their reviews and comments. We are quite happy that the reviewers all fairly unanimously found our work to be interesting and worthwhile. We address each of the reviewers questions and comments in the below.

---

### Decision · Program_Chairs · 2024-09-25

**Decision:**

Accept (spotlight)

**Comment:**

The reviewers were uniformly positive in their assessment of this paper. However, there were consistent concerns with its organization–I encourage the authors to make an effort to organize and polish their paper further, to make it more approachable to readers.